# Maintaining protein stability of ΔNp63 via USP28 is required by squamous cancer cells

Cristian Prieto-Garcia[1,2], Oliver Hartmann[1,2], Michaela Reissland[1,2], Fabian Braun[1,2], Thomas Fischer[1,3], Susanne Walz[4], Christina Schülein-Völk[5], Ursula Eilers[5], Carsten P Ade[2,6], Marco A Calzado[7,8,9], Amir Orian[10], Hans M Maric[11], Christian Münch[12], Mathias Rosenfeldt[2,13], Martin Eilers[2,6] [iD] & Markus E Diefenbacher[1,2,*] [iD]

## Abstract

The transcription factor ΔNp63 is a master regulator of epithelial cell identity and essential for the survival of squamous cell carcinoma (SCC) of lung, head and neck, oesophagus, cervix and skin. Here, we report that the deubiquitylase USP28 stabilizes ΔNp63 and maintains elevated ΔNP63 levels in SCC by counteracting its proteasome-mediated degradation. Impaired USP28 activity, either genetically or pharmacologically, abrogates the transcriptional identity and suppresses growth and survival of human SCC cells. CRISPR/Cas9-engineered *in vivo* mouse models establish that endogenous USP28 is strictly required for both induction and maintenance of lung SCC. Our data strongly suggest that targeting ΔNp63 abundance via inhibition of USP28 is a promising strategy for the treatment of SCC tumours.

**Keywords** ΔNp63; MYC; NOTCH; squamous cell carcinoma; USP28
**Subject Categories** Cancer; Signal Transduction

## Introduction

Each year, around 2 million patients are diagnosed and approximately 1.76 million succumb to lung cancer, making this tumour entity the leading cause of cancer-related death for men and women alike (Bray *et al*, 2018). According to the current WHO classification, lung cancer is classified into two major subtypes depending on marker expression: non-small-cell lung cancer (NSCLC) and small-cell lung cancer (SCLC), causing around 85 or 15% of disease incidences, respectively (Inamura, 2017). NSCLC can be further subdivided, according to marker expression and prevalent mutational aberrations, into adenocarcinomas (ADC) and squamous carcinomas (SCCs). Comprehensive analyses of the mutational landscape show that lung SCC is one of the genetically most complex tumours (Cancer Genome Atlas Research, 2012). As a consequence, little is known about therapeutic targetable vulnerabilities of this disease.

A key regulatory protein in SCC is the p53-related transcription factor ΔNp63, encoded by the *TP63* gene (Su *et al*, 2013). ΔNp63 is highly expressed in lung SCC as well as in SCCs of the skin, head and neck, and oesophagus, in part due to gene amplification (Hibi *et al*, 2000, Tonon *et al*, 2005; Cancer Genome Atlas Research N, 2012). The *TP63* locus encodes multiple mRNAs that give rise to functionally distinct proteins. Notably, transcription from two different promoters produces N-terminal variants either containing or lacking the transactivation domain: TAp63 or ΔNp63 (Deyoung & Ellisen, 2007). The major p63 isoform expressed in squamous epithelium and SCC is ΔNp63α (Rocco *et al*, 2006; Koster *et al*, 2007), which is a master transcription factor that establishes epithelial cell identity, including cytokeratin 5/6 and 14 (Rocco *et al*, 2006; Deyoung & Ellisen, 2007; Su *et al*, 2013; Hamdan & Johnsen, 2018; Somerville *et al*, 2018). In addition, ΔNp63 binds to and thereby inactivates TP53 at promoters of pro-apoptotic genes, suppressing their expression (Westfall *et al*, 2003; Craig *et al*, 2010). ΔNp63 is essential for the survival of skin and pancreatic

1  Department of Biochemistry and Molecular Biology, Protein Stability and Cancer Group, University of Würzburg, Würzburg, Germany
2  Comprehensive Cancer Centre Mainfranken, Würzburg, Germany
3  Department for Radiotherapy, University Hospital Würzburg, Würzburg, Germany
4  Core Unit Bioinformatics, Comprehensive Cancer Centre Mainfranken, University of Würzburg, Würzburg, Germany
5  Core Unit High-Content Microscopy, Biocenter, University of Würzburg, Würzburg, Germany
6  Department of Biochemistry and Molecular Biology, University of Würzburg, Würzburg, Germany
7  Instituto Maimónides de Investigación Biomédica de Córdoba (IMIBIC), Córdoba, Spain
8  Departamento de Biología Celular, Fisiología e Inmunología, Universidad de Córdoba, Córdoba, Spain
9  Hospital Universitario Reina Sofía, Córdoba, Spain
10  Faculty of Medicine, TICC, Technion Haifa, Israel
11  Rudolf-Virchow-Center for Experimental Biomedicine, Würzburg, Germany
12  Institute of Biochemistry II, Goethe University, Frankfurt, Germany
13  Institute for Pathology, University of Würzburg, Würzburg, Germany
 *Corresponding author. Tel: +49 0931 31 88167; Fax: +49 0931 31 84113; E-mail: markus.diefenbacher@uni-wuerzburg.de

SCC cells, since established murine skin SCCs are exquisitely dependent on ΔNp63; acute deletion of *TP63* in advanced, invasive SCC induced rapid and dramatic apoptosis and tumour regression (Rocco *et al*, 2006; Galli *et al*, 2010; Ramsey *et al*, 2013; Su *et al*, 2013; Somerville *et al*, 2018). Collectively, these findings raise the possibility that ΔNp63 is a therapeutic target in SCC tumours.

ΔNp63 is an unstable protein that is continuously turned over by the proteasome upon ubiquitination by E3 ligases, such as the FBXW7 ubiquitin ligase (Galli *et al*, 2010). *FBXW7* is frequently mutated or deleted in SCC tumours (cervix 13.15%, HNSC 7.55%, lung 6.4% and oesophagus 7.29%; cBioPortal, Galli *et al*, 2010; Ruiz *et al*, 2019). Intriguingly, it has been shown previously that the degradation of many targets of FBXW7 is counteracted by the deubiquitylase (DUB) USP28 (Popov *et al*, 2007b). This is in part due to the fact that USP28 exploits binding to FBXW7 to interact with its substrates (Schulein-Volk *et al*, 2014). However, USP28 can also recognize the phosphodegron that is required for the binding of FBXW7 to its substrates in an FBXW7-independent manner (Diefenbacher *et al*, 2015). Loss of USP28 counteracts the loss of Fbxw7 in a murine colon tumour model (Diefenbacher *et al*, 2015; Cremona *et al*, 2016), and acute deletion of USP28 in established tumours increases survival in the APC$^{minΔ/+}$ colorectal tumour model (Diefenbacher *et al*, 2014), while not affecting tissue homeostasis in non-transformed cells (Schulein-Volk *et al*, 2014). Together, these data argue that targeting USP28 may destabilize ΔNp63 and suggest that this strategy may have therapeutic efficacy in SCC.

# Results

### USP28 is highly abundant in human squamous tumours and correlates with poor prognosis

To investigate the mutational as well as the expression status of USP28 in lung cancer, we analysed publicly available datasets of human tumours (Figs 1A and B, and EV1A, B and D). USP28 is rarely lost or mutated, but frequently transcriptionally upregulated in human SCC compared to healthy lung tissue or ADC (adenocarcinoma) patient samples (Figs 1A and B, and EV1A and B). Similarly, the expression of *TP63* was significantly upregulated in SCC samples compared to non-transformed tissue or to ADC samples (Figs 1A and EV1A and B).

Next, we determined the abundance of USP28 protein via immunohistochemistry (IHC) on tissue microarrays and tumour sections of a total of 300 human lung tumour samples. Relative to non-transformed tissue, all samples from different human lung tumour subtypes expressed elevated levels of USP28, with SCC presenting the highest levels (Figs 1C and EV1C), confirming the USP28 mRNA expression data (Fig 1A and B). TP63 protein abundance was evaluated within the same cohort and, like USP28, exhibited the highest protein abundance in SCC tumours compared to ADC and SCLC samples and normal tissue (Figs 1C and EV1C). To evaluate the relevance of both proteins for tumour development, we used publicly available datasets to correlate mRNA expression data with patient survival. Patients with an increased expression of either *ΔNp63* or *USP28* showed a significantly shortened overall survival (Fig 1D). Importantly, this correlation was not a secondary consequence of a generally shorter survival of SCC patients, since USP28 expression correlated with worse prognosis even when only SCC patients were analysed (Fig 1E). Finally, we noted that 3% of lung SCC patients display mutations in *USP28* or a deletion of *USP28,* and those showed a much better disease-free survival compared to USP28 wild-type patients (Fig EV1D). These data indicate that USP28 is upregulated in NSCLC, and high expression of USP28 negatively correlates with overall patient survival in SCC tumours. Additionally, we were able to detect a strong correlation between USP28 and ΔNp63 abundance in lung SCC, indicating a potential crosstalk between both proteins.

### ΔNp63 stability is regulated by USP28 via its catalytic activity

To test whether USP28 controls ΔNp63 protein abundance, we initially expressed HA-tagged USP28 and FLAG-tagged ΔNp63 in HEK293 cells by transient transfection. Immunofluorescence staining using antibodies against USP28 and ΔNp63 revealed that both proteins localize to the nucleus of transfected cells (Appendix Fig S1A). Co-immunoprecipitation experiments showed that ΔNp63 binds to USP28 and *vice versa*, indicating an interaction of both molecules in cells (Fig 2A). Upon co-expression of His-tagged ubiquitin, ΔNp63 was ubiquitylated, as demonstrated by pulldown of His-tagged ubiquitin followed by immunoblot using a ΔNp63-specific antibody, and co-expression of USP28 resulted in the deubiquitylation of ΔNp63 (Fig 2B). To test the chain specificity of substrate deubiquitylation by USP28 on ΔNp63, we ectopically

**Figure 1. USP28 is highly abundant in human squamous tumours and correlates with poor prognosis.**

A   Expression of USP28 (left) and TP63 (right) in human lung squamous cell carcinomas (SCC, *n* = 498), adenocarcinomas (ADC, *n* = 513) and normal non-transformed tissue (normal SCC = 338, normal ADC = 348). Xena UCSC software. In box plots, the centre line reflects the median, the cross represents the mean, and the upper and lower box limits indicate the first and third quartiles. Whiskers extend 1.5× the IQR, and outliers are marked as dots.

B   Correlation of mRNA expression of USP28 and TP63 in lung SCC (left, *n* = 498), ADC (right, *n* = 513) and normal non-transformed tissue (normal SCC = 338, normal ADC = 348). R: Spearman's correlation coefficient; m = Slope. Xena UCSC software.

C   IHC analysis of USP28 and ΔNp63 protein abundance in lung cancer and non-transformed human samples (*n* = 300). The staining intensity was quantified in arbitrary units from 0 up to 3 by three independent pathologists. In box plots, the centre line reflects the median, the cross represents the mean, and the upper and lower box limits indicate the first and third quartiles. Whiskers extend 1.5× the IQR, and outliers are marked as dots. *P*-values were calculated using two-tailed *t*-test.

D   Kaplan–Meier estimator of NSCLC patients stratified by USP28 (left, *n* = 1,145) and TP63 (right, *n* = 1,926) expression. *P*-values were calculated using log-rank test. HR: hazard ratio. KmPlot software.

E   Kaplan–Meier estimator of lung SCC patients stratified by USP28 expression (*n* = 271). The *P*-value was calculated using a log-rank test. HR: hazard ratio. KmPlot software.

Data information: *$P < 0.05$; **$P < 0.01$; ***$P < 0.001$; ****$P < 0.001$. See also Fig EV1 and Appendix Table S3 (exact *P*-values and statistical test used).

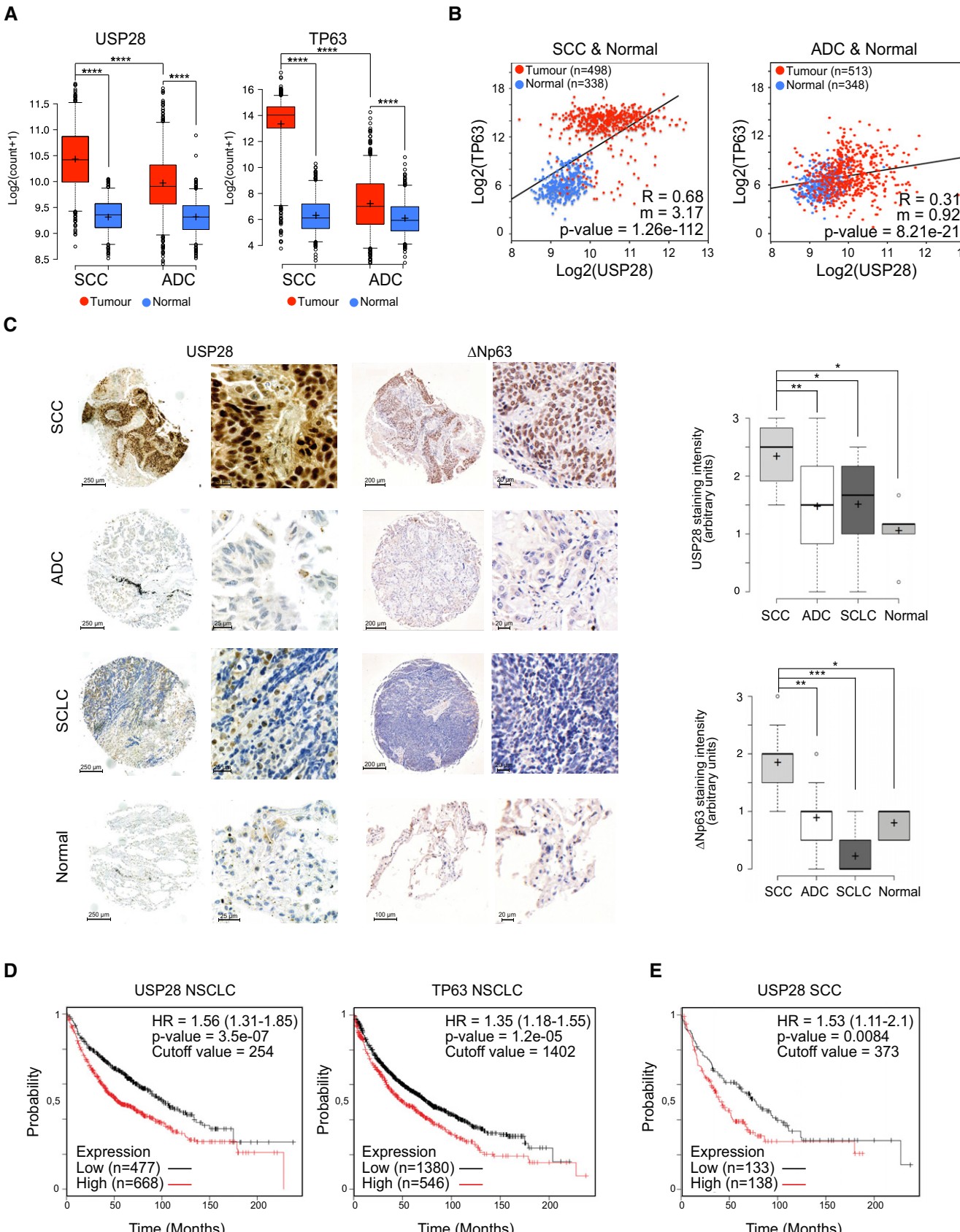

Figure 1.

co-expressed a His-tagged ubiquitin that carries a single lysine residue either at position K48 or K63. Upon His-ubiquitin pulldown, K48- as well as K63-linked poly-ubiquitin chains could be detected on ΔNp63, as previously reported (Galli *et al*, 2010; Peschiaroli *et al*, 2010; Fig 2C). Upon overexpression of USP28, only K48-linked ubiquitin chains were removed from ΔNp63, whereas K63-linked chains were resistant to USP28 (Fig 2C). To test whether USP28 catalytic activity is required for deubiquitination of ΔNp63, we used a catalytic inactive mutant of USP28, USP28$^{C171A}$ (Fig 2D–F; Popov *et al*, 2007b; Diefenbacher *et al*, 2014, 2015; Schulein-Volk *et al*, 2014). Immunoprecipitation of transfected cells using an ΔNp63-specific antibody revealed that USP28$^{C171A}$ was able to bind to ΔNp63 (Fig 2D). While overexpression of the wild-type form of USP28 deubiquitylated ΔNp63 (Fig 2E), USP28$^{C171A}$ failed to do so, demonstrating that the catalytically active cysteine of USP28 is required for deubiquitylation of ΔNp63 (Fig 2E).

K48-linked ubiquitin chains target proteins to the proteasome for degradation (Grice & Nathan, 2016). Since USP28 is able to counter-act K48-linked ubiquitylation of ΔNp63, we investigated the ability of USP28 to modulate ΔNp63 protein turnover. To do so, we co-expressed ΔNp63 with either wild-type USP28 or USP28$^{C171A}$ in HEK293 cells. Twenty-four hours post-transfection, cells were treated with 100 µg/ml cycloheximide (CHX) to block protein synthesis. Co-expression of wild-type USP28, but not of the catalytically inactive mutant, strongly stabilized ΔNp63 protein (Fig 2F).

As ΔNp63 protein stability was enhanced by USP28, but not via the catalytic inactive C171A mutant, we tested whether a pharmacologic inhibitor of DUBs, PR-619, would also affect overall protein abundance of ΔNp63. Therefore, we expressed ΔNp63 in HEK293 cells and, 24 h post-transfection, treated cells with either DMSO or increasing amounts of PR-619 for additional 24 h (Fig 2G). While ΔNp63 was not degraded in control cells treated with DMSO, cells exposed to PR-619 showed a shortened half-life of 8 h for ΔNp63 protein (PR-619 IC$_{50}$ of < 5 µM, Fig 2G). In control-treated cells, the protein abundance of USP28 was not affected; however, upon addition of the pan-DUB inhibitor PR-619, USP28 protein was reduced in

a dose-dependent fashion (Fig 2G). This is in line with previous observations that the enzymatic activity of DUBs is required to enhance their own stability (de Bie & Ciechanover, 2011; Wang *et al*, 2017). Collectively, these data demonstrate that USP28 can interact with and stabilize the ΔNp63 protein by removing K48-linked ubiquitin chains and that the catalytic domain of USP28 is required for this activity.

### USP28 stabilizes ΔNp63 independently of FBXW7

Previous reports highlighted the regulation of ΔNp63 protein stability by the E3 ligase FBXW7 (Galli *et al*, 2010), which is commonly mutated or lost in human SCC of various origins (Appendix Fig S2A and B). To identify via which protein domain ΔNp63 interacts with USP28, we performed peptide spot interaction studies (Appendix Fig S1B, Materials and Methods) and were able to identify, apart from several lysine-containing domains, the Fbxw7 phosphodegron as a putative interaction site for USP28 (Appendix Fig S1B). To investigate whether USP28 interacts with ΔNp63 in a FBXW7-dependent fashion and whether the phosphodegron motive is required to facilitate the interaction, we made use of a ΔNp63 point mutant, ΔNp63$^{S383A}$, which is not phosphorylated by GSK3β and abolishes binding to FBXW7 (Galli *et al*, 2010). Ectopic expression of USP28 and ΔNp63$^{S383A}$ in HEK293 cells showed that USP28 was able to increase ΔNp63$^{S383A}$ abundance (Appendix Fig S1C). Furthermore, by co-immunoprecipitation experiments with exogenous USP28 and ΔNp63$^{S383A}$ in HEK293 cells, we were able to detect that USP28 binds to ΔNp63$^{S383A}$ (Appendix Fig S1D), This interaction resulted in a decreased ubiquitylation of ΔNp63$^{S383A}$ (Appendix Fig S1E). Furthermore, overexpression of USP28 was able to increase protein half-life (Appendix Fig S1F) and treatment of cells with PR-619 affected ΔNp6$^{3S383A}$ protein stability, albeit to a somewhat lesser extent compared to wild-type ΔNp63 (Appendix Fig S1G). To determine whether endogenous USP28 regulates the abundance and stability of ΔNp63, we used a human SCC cell line (A-431). These cells are homozygous for the S462Y mutation in FBXW7, which is

**Figure 2. ΔNp63 stability is regulated by USP28 via its catalytic activity.**

A    Co-immunoprecipitation of exogenous HA-USP28 and FLAG-ΔNp63 in HEK293 cells. Either HA-USP28 or FLAG-ΔNp63 were precipitated and blotted against FLAG-ΔNp63 or HA-USP28. The input corresponds to 10% of the total protein amount used for the IP (ACTIN as loading control).

B    Ni-NTA His-ubiquitin pulldown in control-transfected or HA-USP28-overexpressing HEK293 cells, followed by immunoblot against exogenous ΔNp63. The input corresponds to 10% of the total protein amount used for the pulldown. Relative ubiquitination of the representative immunoblot was calculated using ACTIN for normalization.

C    Ni-NTA His-ubiquitin pulldown K48 or K63 in control and HA-USP28-overexpressing HEK293 cells, followed by immunoblot against exogenous ΔNp63. The input corresponds to 10% of the total protein amount used for the pulldown. Relative ubiquitination of the representative immunoblot was calculated using VINCULIN for normalization.

D    Co-immunoprecipitation of exogenous FLAG-USP28 C171A and FLAG-ΔNp63 in HEK293 cells. ΔNp63 was precipitated and blotted against FLAG-USP28 or ΔNp63. The input corresponds to 10% of the total protein amount used for the IP (ACTIN as loading control).

E    Ni-NTA His-ubiquitin pulldown in control-, FLAG-USP28- or FLAG-USP28 C171A-transfected HEK293 cells, followed by immunoblot against exogenous ΔNp63. The input corresponds to 10% of the total protein amount used for the pulldown. Relative ubiquitination of the representative immunoblot was calculated using ACTIN for normalization.

F    CHX chase assay (100 µg/ml) of control-, FLAG-USP28- or FLAG-USP28 C171A-transfected HEK293 cells for indicated time points. Representative immunoblot analysis of FLAG (USP28) and ΔNp63 as well as quantification of relative protein abundance (ACTIN as loading control).

G    Immunoblot of USP28 and ΔNp63 in transfected HEK293 cells upon treatment with either DMSO or indicated concentrations of PR-619 for 24 h. Relative protein abundance was calculated ACTIN as loading control.

Data information: Western blots shown are representative of three independent experiments (*n* = 3). All quantitative graphs are represented as mean ± SD of three experiments (*n* = 3). *P*-values were calculated using two-tailed *t*-test statistical analysis; *P* < 0.05; **P* < 0.01; see also Appendix Fig S1 and Appendix Table S3 (exact *P*-values and statistical test used).

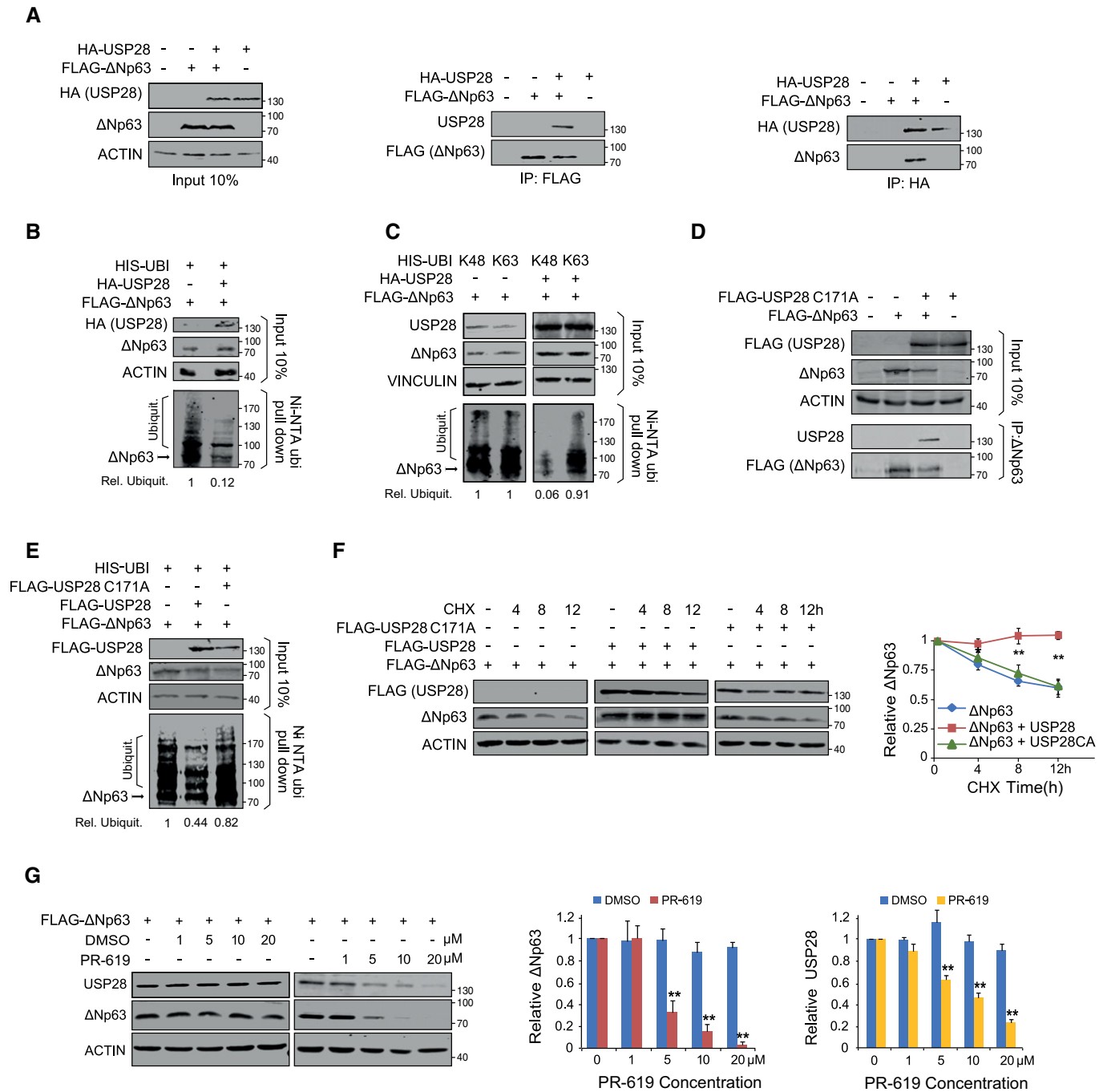

Figure 2.

thought to inactivate substrate recognition (Appendix Fig S2C and D; Yeh et al, 2016). FBXW7, USP28 and ΔNp63 were readily detectable in the nucleus of these cells by immunofluorescence (Appendix Fig S2C and E). Furthermore, immunoprecipitation of endogenous USP28 co-immunoprecipitated endogenous ΔNp63, and vice versa (Fig 3A). In contrast, antibodies against USP25, a ubiquitin-specific protease that is structurally very similar to USP28 (Appendix Fig S2F; Gersch et al, 2019; Sauer et al, 2019), did not co-immunoprecipitate ΔNp63 although USP25 is readily detectable in A-431 cells. Correspondingly, antibodies against ΔNp63 did not

co-immunoprecipitate endogenous USP25 (Fig 3A). Next, we tested the dependence of DUB-mediated protein stability on endogenous ΔNp63 by using PR-619. A-431 cells were treated either with DMSO or increasing amounts of PR-619 for 24 h (Appendix Fig S2G). Cells exposed to PR-619 showed loss of endogenous ΔNp63. This degradation was mediated via the 26s proteasome, as addition of MG132 restored protein levels of ΔNp63 and USP28 (Appendix Fig S2G).

To investigate whether USP28 regulates ΔNp63 protein stability in A-431 cells, we generated cell lines expressing either a

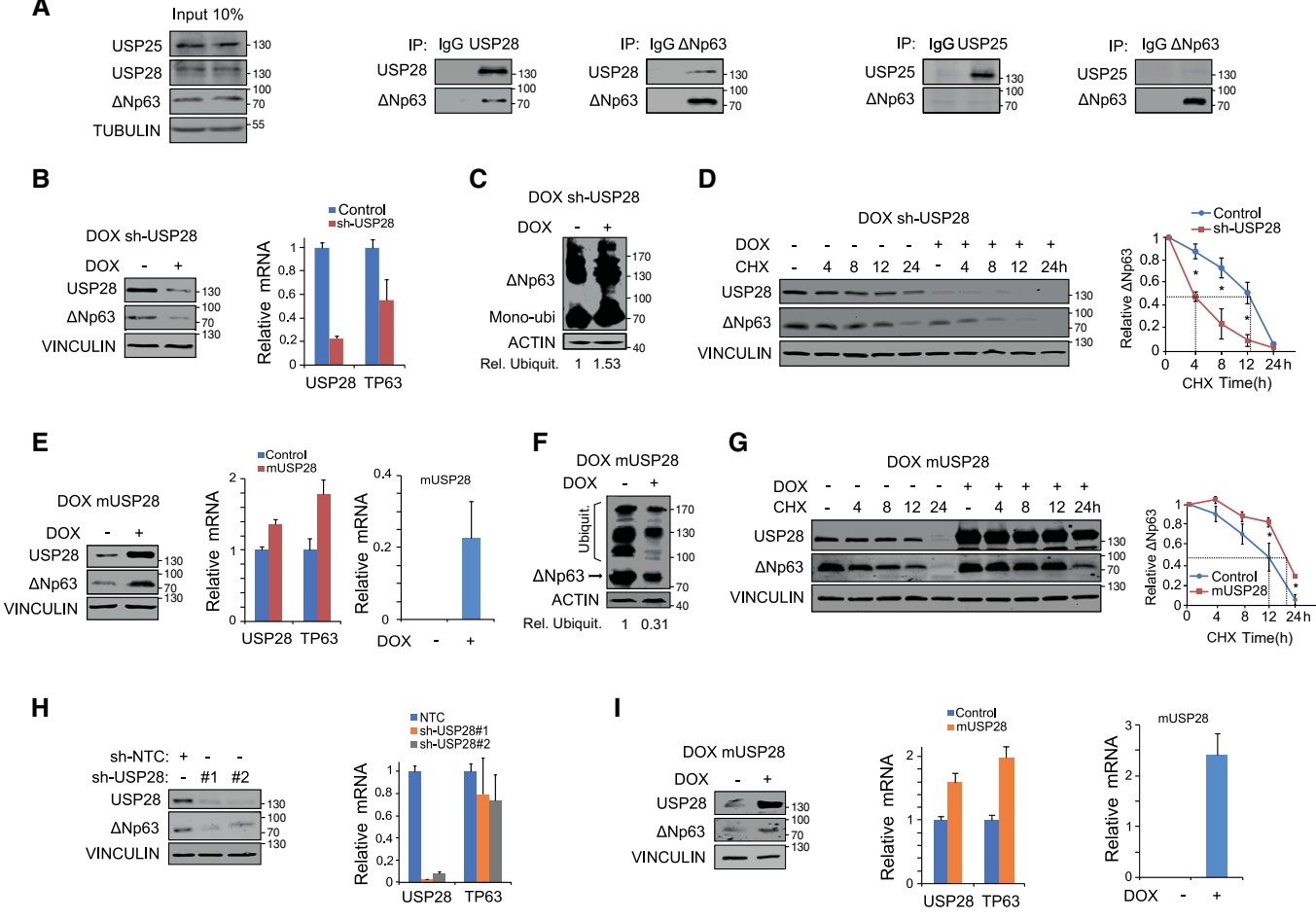

**Figure 3. USP28 regulates ΔNp63 protein stability in SCC tumour cell lines.**

A   Immunoblot of endogenous USP25, USP28 and ΔNp63 immunoprecipitated (IP) from A-431 cells and co-precipitated ΔNp63 or USP28. Beads were coupled to specific USP25, USP28 and ΔNp63 antibodies or non-specific rabbit IgG as control. The input corresponds to 10% of the total protein amount used for the IP (TUBULIN as loading control).

B   Inducible depletion of USP28 in A-431 upon treatment with doxycycline (1 μg/ml) for 96 h, Western blot (left, VINCULIN as loading control) and qPCR analysis of USP28 and ΔNp63 expression relative to ACTIN (right) were performed.

C   Tandem ubiquitin binding entity (TUBE) pulldown of endogenous ubiquitylated ΔNp63 in A-431 cells upon DOX depletion of USP28. Relative ubiquitination of the representative immunoblot was calculated using ACTIN for normalization.

D   Cycloheximide (CHX) chase assay (100 μg/ml) of control or inducible sh-USP28 A-431 cell line (EtOH or 1 μg/ml dox) for indicated time points. Representative immunoblot (left, VINCULIN as loading control) of USP28 and ΔNp63 and quantification of relative protein abundance (right).

E   Doxycycline induced murine USP28 overexpression (EtOH or 1 μg/ml dox for 96 h) in A-431 cells followed by immunoblot (VINCULIN as loading control) and qPCR analysis of USP28 and ΔNp63. For qPCR, human USP28 and murine USP28 (mUSP28) primers were used. Relative mRNA was calculated using ΔΔCt analysis for human USP28 and ΔCt for mUSP28 (ACTIN as housekeeping).

F   TUBE pulldown of endogenous ubiquitylated ΔNp63 in A-431 cells upon overexpression of mUSP28 for 96 h (EtOH or 1 μg/ml dox). Relative ubiquitination of the representative immunoblot was calculated using ACTIN for normalization.

G   CHX chase assay (100 μg/ml) of control or inducible mUSP28-overexpressing A-431 cell line (EtOH or 1 μg/ml dox) for indicated time points. Representative immunoblot analysis of USP28 and ΔNp63 as well as quantification of relative protein abundance (VINCULIN as loading control).

H   Immunoblot of control (sh-NTC) and two independent shRNA targeting USP28 (sh-USP28#1 and #2) for ΔNp63 and USP28 protein abundance in LUDLU-1[adh] (VINCULIN as loading control), followed by qPCR analysis of USP28 and ΔNp63 expression relative to ACTIN.

I   Doxycycline induced mUSP28 overexpression (EtOH or 1 μg/ml dox for 96 h) in LUDLU-1[adh] cells followed by immunoblot (VINCULIN as loading control) and qPCR analysis of USP28, mUSP28 and ΔNp63. Relative mRNA was calculated using ΔΔCt analysis for human USP28 and ΔCt for mUSP28 (ACTIN as housekeeping for the analysis).

Data information: Western blots shown are representative of three independent experiments ($n = 3$). All quantitative graphs are represented as mean ± SD of three experiments ($n = 3$). $P$-values were calculated using two-tailed $t$-test statistical analysis; *$P < 0.05$; see also Appendix Fig S2 and Appendix Table S3 (exact $P$-values and statistical test used).

Source data are available online for this figure.

doxycycline-inducible or constitutive shRNA targeting USP28 (Figs 3B and EV2D, Appendix Fig S2H) and investigated the effects of acute USP28 depletion on ΔNp63 protein. ΔNp63 protein abundance was reduced in USP28-depleted cells (Figs 3B and EV2D, Appendix Fig S2H) as detected by Western blot. Notably, depletion of USP28 also reduced ΔNp63 mRNA levels, consistent with previous observations that ΔNp63 activates the expression of its own mRNA (Antonini *et al*, 2006) (Fig 3B). To assess the impact of acute USP28 depletion on endogenous ΔNp63 ubiquity-lation, we performed tandem ubiquitin binding entity (TUBE) pulldown assays (Hjerpe *et al*, 2009). TUBEs are composed of four copies of the ubiquitin-associated domain of ubiquilin fused in tandem to a glutathione S-transferase (GST) tag and enable the detection of endogenous ubiquitin modifications on target proteins. TUBE ubiquitin pulldown experiments with cell lines expressing inducible shRNA targeting USP28 revealed increased ubiquitylation of ΔNp63 upon reduction of USP28 (Fig 3C and Appendix Fig S2I). Next, we asked whether acute loss of USP28 affects ΔNp63 protein half-life. Inducible shRNA cell lines were cultured in the presence of EtOH (control) or doxycycline (1 μg/ml) for 72 h prior to CHX treatment for the indicated time points, and USP28 and ΔNp63 protein abundance was measured by Western blot. Acute loss of USP28 reduced ΔNp63 protein abundance to 50% within 4 h, compared to a half-life of 12 h observed in control cells (Fig 3D).

To test whether increasing the levels of USP28 results in an increased protein stability of ΔNp63 in a SCC cell line, we generated a doxycycline-inducible system for the overexpression of murine USP28 in A-431 cells (Fig 3E–G), as murine USP28 is structurally very similar to human USP28 (Appendix Fig S2F). Upon treatment with doxycycline, elevated amounts of USP28 resulted in an increase in ΔNp63 protein levels (Fig 3E). The effects on ΔNp63 were not only detectable on protein level, but were also reflected in increased levels of ΔNp63 mRNA, most likely due to the positive autoregulation of p63 transcription discussed before (Fig 3E). Exogenous USP28 also resulted in the decreased ubiquitylation of ΔNp63 as detected by TUBE assay (Fig 3F) and enhanced protein stability, extending its half-life from 12 h to around 20 h (Fig 3G). Conversely, overexpression of the catalytic inactive USP28[C171A] in A-431 failed to stabilize endogenous ΔNp63 (Appendix Fig S2J) and instead resulted in an increase in ΔNp63 ubiquitylation, as measured by TUBE assay (Appendix Fig S2K). This shows the ability of USP28 to deubiquitylate ΔNp63 and increase the protein stability and demonstrates that the catalytically active cysteine of USP28 is required for this function.

The observed regulation of ΔNp63 in A-431 cells argues that USP28 stabilizes ΔNp63 in an FBXW7-independent manner. To confirm that USP28 can also regulate ΔNp63 in FBXW7 wild-type SCC cells, we depleted USP28 using two independent shRNAs in LUDLU-1[adh] cells (Fig 3H). Similar to the results obtained in A-431 cells, depletion of USP28 resulted in the reduction of ΔNp63 protein abundance (Fig 3H). Furthermore, doxycycline-inducible overexpression of murine USP28 further increased endogenous ΔNp63 protein levels (Fig 3I) and enhanced ΔNp63 mRNA levels, highlighting a positive feedback loop for ΔNp63.

These data demonstrate that USP28 deubiquitylates and stabilizes ΔNp63 and that both substrate recognition and stabilization occur in an FBXW7-independent manner.

## The ΔNp63-USP28 axis is required to maintain the identity of SCC cells

Both SCC cells and tumours depend on ΔNp63 for maintaining proliferation and cell identity by maintaining the expression of lineage markers of keratinization, such as keratins 5 and 14, which are not found in ADC (McDade *et al*, 2011; Lau *et al*, 2013; Ramsey *et al*, 2013). To explore whether USP28 controls these biological functions of ΔNp63, we first targeted ΔNp63 by two independent shRNA sequences and analysed the knock-down efficacy by immunoblotting (Fig EV2A). Both shRNAs led to a significant decrease in ΔNp63 protein levels (Fig EV2A). Control or ΔNp63-depleted A-431 cells were seeded at equal cell density and counted at indicated time points (Fig EV2B). Depletion of ΔNp63 decreased SCC proliferation (Fig EV2B) and cell cycle profiling indicated a mild accumulation of cells in S-phase (Fig EV2C), consistent with previous reports (Wang *et al*, 2019). Depletion of USP28 using two independent shRNAs, which decreased ΔNp63 levels by at least 70%, caused a very similar decrease in cell proliferation and cell cycle progression (Fig EV2D–F).

To investigate whether USP28, like ΔNp63, is required to maintain the characteristic gene expression pattern of SCC cells, RNA expression profiles of A-431 cells stably expressing shRNAs targeting either USP28 or ΔNp63 were compared to cells expressing a non-targeting control shRNA. Analysis of global gene expression revealed a strong correlation of target gene regulation in response to depletion of USP28 or ΔNp63 ($R = 0.48$, m = 0.72, $P < 2.2e$-308, Fig 4A). There were 266 commonly downregulated genes and 66 commonly upregulated genes (Fig 4B and C). Gene set enrichment analysis (GSEA) using genes downregulated and upregulated in response to depletion of either USP28 or ΔNp63 (Appendix Table S1), respectively, confirmed the strong similarity in expression changes caused by depletion of either factor (Figs 4D and E, and EV2G and H).

To gain insight into the biological processes that underlie this similarity, we performed GO term analysis and found that depletion of either ΔNp63 or USP28 strongly downregulates genes mapping to a set of overlapping GO terms that describe epithelial cell identity and keratin expression (Fig 4F and G). As SCC depends on the expression of ΔNp63 to maintain proliferation (Fig EV2A and B), we wondered if overexpression of ΔNp63 is able to rescue the observed proliferative defect in USP28 knock-down A-431 cells. While depletion of USP28 reduced ΔNp63 and KRT14 (SCC marker) protein abundance, exogenous ΔNp63 restored KRT14 expression in USP28 knock-down cells and partially restored proliferation (Fig 4H and I). These data indicate that depletion of USP28 affected cellular proliferation in SCC via reducing ΔNp63 levels.

To determine whether USP28 regulates genes involved in epithelial cell identity and SCC tumour formation via ΔNp63, we tested whether ectopic expression of ΔNp63 restores the expression of these genes in USP28-depleted cells. RT–PCR analyses showed that depletion of ΔNp63 or USP28 decreases expression of keratins KRT5, KRT14 and KRT19 in A-431 cells and expression of exogenous ΔNp63 partially restored expression of these SCC cancer genes in USP28-depleted cells (Fig 4J), demonstrating that the effect of USP28 is mediated in part by the downregulation of ΔNp63. In contrast to SCC-associated cytokeratins, KRT10, which is a marker of differentiation (Saladi *et al*, 2017), was upregulated

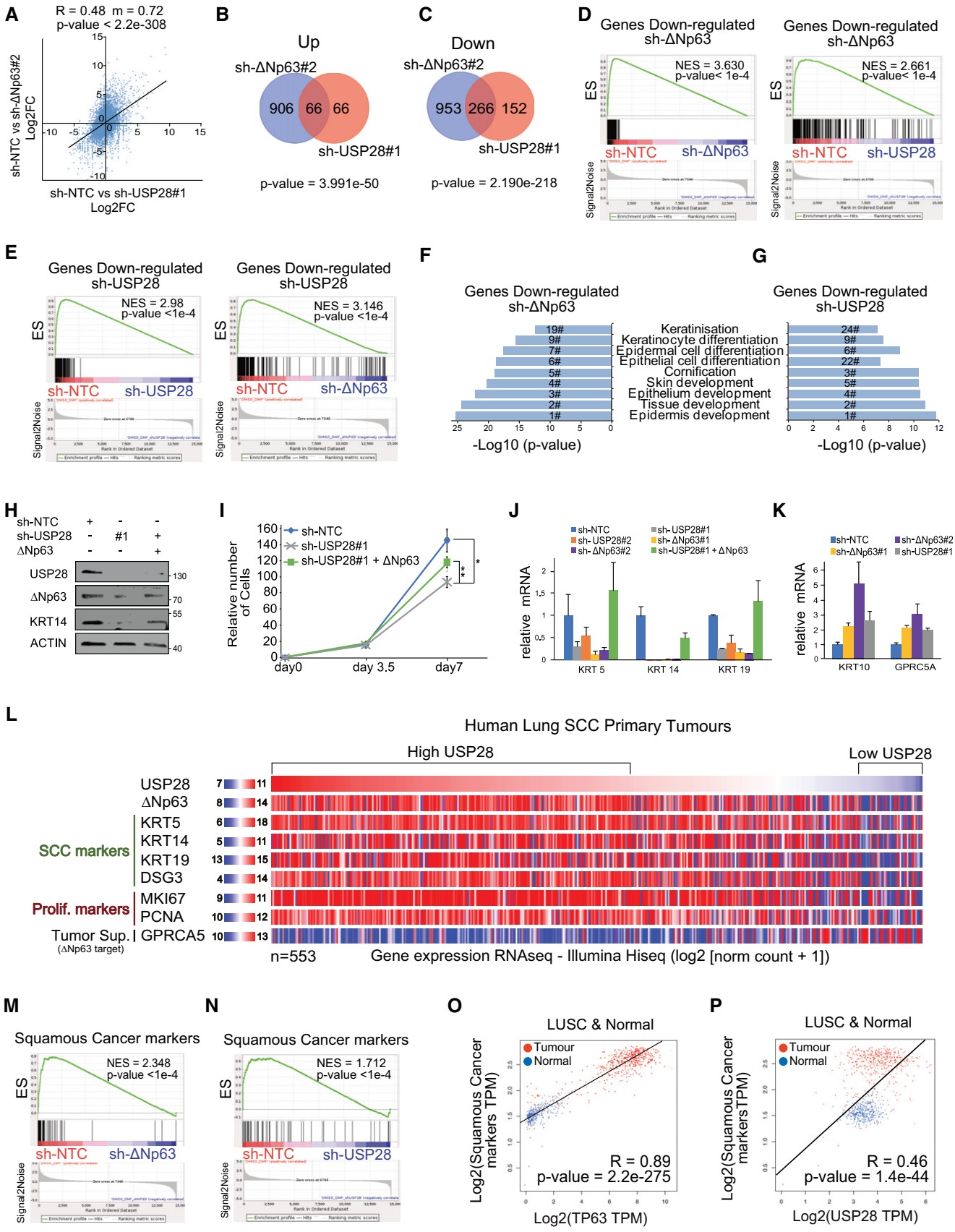

Figure 4.

◀

**Figure 4.  SCC tumour cells are dependent on USP28 and/or ΔNp63 to maintain a SCC identity.**

A   Correlation of gene expression changes upon constitutive transduction of A-431 cells with either shRNA targeting USP28 (sh-USP28#1) or ΔNp63 (sh-ΔNp63#2) relative to non-targeting control (sh-NTC). The diagonal line reflects a regression build on a linear model. R: Pearson's correlation coefficient, m: slope of the linear regression model.

B   Venn diagram of differentially upregulated genes ($\log_2$FC > 1.5 and $q$ < 0.05) between sh-ΔNp63#2 and sh-USP28#1 relative to non-targeting control (sh-NTC) in A-431 cells. $P$-values were calculated using a hypergeometric test.

C   Venn diagram of differentially downregulated genes ($\log_2$FC < 1.5 and $q$ < 0.05) between sh-ΔNp63#2 and sh-USP28#1 relative to sh-NTC in A-431 cells. $P$-values were calculated using a hypergeometric test.

D   Gene set enrichment analysis (GSEA) of a gene set of significantly downregulated genes in sh-ΔNp63#2-transfected A-431 cells ("Down-regulated sh-ΔNP63", Appendix Table S1). The gene set was analysed in sh-ΔNp63#2- (left) and sh-USP28#1-depleted (right) A-431 cells. (N)ES: (normalized) enrichment score.

E   Gene set enrichment analysis (GSEA) of a gene set of significantly downregulated genes in sh-USP28#1-transfected A-431 cells ("Down-regulated sh-USP28", Appendix Table S1). The gene set was analysed in shUSP28#1- (left) and sh-ΔNp63#-depleted (right) A-431 cells. (N)ES: (normalized) enrichment score.

F   GO term analysis of biological processes enriched in sh-ΔNp63#2-depleted A-431 cells relative to sh-NTC. Numbers indicate the ranking position of all analysed GO terms based on the significance.

G   GO term analysis of biological processes enriched in sh-USP28#1-depleted A-431 cells relative to sh-NTC. Numbers indicate the ranking position of all analysed GO terms based on the significance.

H   Immunoblot of endogenous ΔNp63 and USP28 in A-431 cells stably transduced with constitutive shRNA-non-targeting control (NTC) or against USP28 and transiently transfected with exogenous ΔNp63. ACTIN served as loading control. Representative Western blot from three independent experiments.

I   Cell growth of A-431 cells stably transduced with constitutive shRNA-non-targeting control (NTC) or against USP28 and transiently transfected with exogenous ΔNp63. Total cell number was measured and assessed at indicated time points. Quantitative graph is represented as mean ± SD of three experiments ($n$ = 3). $P$-values were calculated using two-tailed $t$-test statistical analysis. *$P$ < 0.05, **$P$ < 0.01.

J   Relative expression of SCC marker genes KRT5, KRT14 and KRT19 in A-431 cells stably transduced with constitutive shRNA-non-targeting control (NTC), two independent shRNA-ΔNp63, two independent shRNA-USP28 or ΔNp63 in shRNA-USP28#1, normalized to ACTIN. Quantitative graph is represented as mean ± SD of three experiments ($n$ = 3).

K   Relative expression of epithelial marker genes KRT10 and GPCR5A in A-431 cells stably transduced with constitutive shRNA-non-targeting control (NTC), two independent shRNA-ΔNp63 and shRNA-USP28#1, normalized to ACTIN. Quantitative graph is represented as mean ± SD of three experiments ($n$ = 3).

L   Genomic signature of primary human lung SCC samples comprising USP28, ΔNp63, KRT5, KRT14, KRT19, DSG3, MKI67, PCNA and GPRC5A. Samples were sorted dependent on relative USP28 expression (high to low). $n$ = 553. Xena UCSC software.

M   GSEA of consensus squamous cancer marker genes (see Appendix Table S2) in A-431 cells stably transduced with sh-ΔNp63#2 or sh-NTC. (N)ES: (normalized) enrichment score.

N   GSEA of consensus squamous cancer marker genes in A-431 cells stably transduced with sh-USP28#1 or sh-NTC. (N)ES: (normalized) enrichment score.

O   Correlation of mRNA expression of consensus squamous cancer marker genes and TP63 in lung SCC and non-transformed lung tissue (Normal). R: Spearman's correlation coefficient. $n$ = 836. GEPIA software.

P   Correlation of mRNA expression of consensus squamous cancer marker genes and USP28 in lung SCC and non-transformed lung tissue (Normal). R: Spearman's correlation coefficient. $N$ = 836. GEPIA software.

Data information: See also Fig EV2 and Appendix Table S3 (exact $P$-values and statistical test used).

in ΔNp63- and USP28-depleted A-431 (Fig 4K). Similar responses were observed for the putative tumour suppressor GPRC5A (Saladi *et al*, 2017); loss of either protein resulted in an increased expression (Fig 4K), thereby contributing to cellular differentiation and suppression of colony formation. Hence, loss of USP28 or ΔNp63 changes the cellular fate and signature of SCC tumour cells. By analysing publicly available RNA-sequencing data of primary human lung SCC, we could observe a similar correlation between USP28, ΔNp63 and SCC marker gene expression, such as KRT5, KRT14 and KRT19 and the tumour suppressor GPRC5A (Fig 4L).

Since the transcriptional effects observed in the knock-down cell lines by GO term analysis pointed towards pathways frequently found deregulated in SCC tumours (Rocco *et al*, 2006; Koster *et al*, 2007), which was also observed by the strong effects on KRT5, KRT14 and KRT19 expression (Fig 4H, J and L), we next wanted to know whether other SCC markers are also regulated by the USP28/ΔNp63 axis. Therefore, we analysed a panel of SCC-relevant marker genes, which have previously been shown to be expressed in all SCC subclasses (Wilkerson *et al*, 2010; Mukhopadhyay *et al*, 2014; Xu *et al*, 2014a; Ferone *et al*, 2016; Fig EV2I, Appendix Table S2). We analysed the expression of these genes in control, ΔNp63 and USP28 knock-down A-431 cells. In cells depleted of ΔNp63, genes associated with SCC were downregulated (Fig 4M), in line with the function of ΔNp63 as a master regulator of SCC identity. Analysing

the SCC marker gene panel in USP28 knock-down cells (Fig 4N) also revealed a striking similarity to ΔNp63 knock-down (Fig 4M). Furthermore, SCC markers were commonly downregulated in cells with reduced amounts of USP28 (Fig 4N). To assess whether the observed genetic alterations are attributed to the USP28-ΔNp63 axis in SCC, we next analysed "Hallmark" gene sets of reported USP28 substrates NOTCH1, MYC and AP1-c-JUN, in USP28 and ΔNp63 knock-down A-431 cells (Fig EV2J and K). In neither case, a significant downregulation, as seen for ΔNp63 (Fig 4D), could be observed, indicating that the biological effects observed on SCC cell identity are specific to the regulation of ΔNp63 by USP28. Additionally, no compensatory gene expression mechanism was observed in ΔNp63-silenced cells (Fig EV2K), demonstrating that ΔNp63 is a vulnerability of SCC.

Furthermore, publicly available expression datasets highlight the strong correlation of ΔNp63 and SCC marker co-expression in human tumour samples (Spearman $R = 0.89$, $P = 2.2e-275$; Fig 4O) and, in line with the observed dependency of SCC cells for USP28, a correlation between SCC marker expression and USP28 abundance (Spearman $R = 0.46$, $P = 1.4e-44$; Fig 4P).

Collectively, the data show that USP28 regulates characteristic SCC gene expression profiles via its ability to control ΔNp63 protein abundance. Hence, cells depleted for USP28 or ΔNp63 show a partial but essential overlap in SCC-specific genes. Therefore, we were able to show that the USP28-ΔNp63 axis is implicated in the

regulation and control of the expression of SCC marker genes as well as genes controlling cell fate and SCC identity.

## SCC depends on USP28 expression for tumour induction and engraftment

We next used genetic tools to interrogate the role of USP28 in induction of SCC. To ablate USP28 during tumour initiation, we used CRISPR/Cas9-mediated gene targeting. To induce primary lesions in the lungs of mice, we used a constitutive $Rosa26^{Sor}$-CAGG-Cas9-IRES-GFP transgenic mouse strain and intratracheally infected these mice at 8 weeks of age with adeno-associated virus (AAV) virions containing sgRNA cassettes targeting sequences that inactivate Tp53 ($p53^{\Delta}$) and Stk11/Lkb1 ($Lkb1^{\Delta}$) and introduce the oncogenic mutation G12D, via a repair template, into the $KRas$ locus. We refer to these mice as KP ($Kras^{G12D}:Tp53^{\Delta}$) or KPL ($Kras^{G12D}; Tp53^{\Delta}; Lkb1^{\Delta}$). At 12 weeks post-intratracheal intubation, KP mice developed ADC tumours as determined by the expression of the ADC marker TTF1 and the absence of the SCC marker KRT5 (Fig EV3A–C). Co-depletion of the tumour suppressor $STK11/Lkb1$, in combination with $Tp53$ and $KRas$ targeting, resulted in the development of both major NSCLC entities, ADC (TTF1$^{+}$/$\Delta$Np63$^{-}$/KRT5$^{-}$) and SCC (TTF1$^{-}$/$\Delta$Np63$^{+}$/KRT5$^{+}$; Fig 5A–C). Loss of $Stk11/Lkb1$ in KPL mice dramatically increased tumour area and shortened overall survival compared to that of KP mice (Fig EV3D and E). Evaluation of USP28 abundance, estimated by IHC, demonstrated an increase in USP28 protein in SCC tumours compared to ADC tumours within the same KPL animal (Fig 5C).

To test the role of USP28 in tumour induction, we included two sgRNA cassettes targeting USP28 into the experimental cohort of KPL mice ($USP28^{\Delta}$, referred to as KPLU). Concomitant targeting of USP28 at the time of tumour induction significantly affected NSCLC formation (Fig 5B). Both total tumour area and tumour number per animal were significantly reduced in KPLU compared to KPL mice (Fig 5D and E). Analysis of present tumour types using IHC staining for marker proteins of ADC and SCC revealed that loss of USP28 completely abolished the presence of SCC and negatively affected the abundance of ADC in KPLU mice (Figs 5C and EV3F).

Tumours developing in KPL mice showed expression of USP28 and $\Delta$Np63, while USP28 was strongly reduced in isolated tumours from KPLU mice and $\Delta$Np63 was not detectable (Fig 5C). Immunoblotting of macroscopically excised primary tumours showed the lack of expression of $\Delta$Np63 in KPLU tumours, when compared to samples obtained from KPL mice (Fig 5F). It is noteworthy that the USP28 targets c-JUN and c-MYC were not downregulated in established tumours in KPLU mice, when compared to KPL. However, active NOTCH1 was reduced, in protein abundance and IHC staining intensity upon targeting of USP28 (Figs 5G and EV3G). This is in line with the observed reduction of Notch1 signature genes upon genetic depletion of USP28 or $\Delta$Np63 in A-431 cells (Fig EV2J and K). Consistent with this effect, the survival of KPLU mice was significantly prolonged compared to KPL mice (Fig 5H). The observed effects of USP28 depletion during tumour induction suggest that USP28 is required for SCC formation $in\ vivo$.

In a second series of experiments, we assessed a potential role of USP28 in tumour engraftment. To do so, we made use of a genetically modified mouse model (GEMM) harbouring conditional alleles for Tp53 ($Tp53^{flox/flox}$) and a Cre-activatable mutant KRas ($lsl$-$KRasG12D$). We intratracheally administered AAV encoding Cre recombinase, packaged with the capsid 2/8, which facilitates a tropism towards tracheal and alveolar cells (Fig EV3H; Winters $et\ al$, 2017). Twelve weeks post-infection, mice were sacrificed and

---

**Figure 5. SCC depends on USP28 expression for tumour induction and maintenance.**

A  Schematic diagram of CRISPR/Cas9-mediated tumour modelling and targeting of $p53^{\Delta}$; $Lkb1^{\Delta}$; $KRas^{G12D}$(KPL) or $Usp28^{\Delta}$; $p53^{\Delta}$; $Lkb1^{\Delta}$:$KRas^{G12D}$(KPLU) mouse lines in $Rosa26Sor$-CAGG-Cas9-IRES-GFP mice.

B  Representative haematoxylin and eosin (H&E) staining of tumour-bearing animals 12 weeks post-intratracheal infection. Boxes indicate highlighted tumour areas in (C) (a, b, a' and b'). Scale bar = 2,000 μm; nKPL = 6 and nKPLU = 5. H = heart.

C  Representative IHC staining for ADC (TTF-1) and SCC (KRT5 and $\Delta$Np63) marker expression as well as Usp28 abundance in KPL ($n$ = 6) and KPLU ($n$ = 5) lung tumours. Scale bar: 20 μm.

D  Quantification of % tumour area (normalized to total lung area) in KPL ($n$ = 6) and KPLU ($n$ = 5) animals.

E  Quantification of lung tumour numbers in KPL ($n$ = 6) and KPLU ($n$ = 5) animals.

F  Immunoblot of endogenous LKB1, USP28 and $\Delta$Np63 in KPL and KPLU tumours (VINCULIN as loading control). Western blot shown is representative of three independent experiments ($n$ = 3).

G  Representative immunoblot of endogenous NOTCH1, c-MYC and c-JUN in KPL and KPLU (ACTIN as a control). $n$ = 3

H  Kaplan–Meier survival curves comparing KPL ($n$ = 6) and KPLU ($n$ = 5) animals upon AAV intratracheal infection. The $P$-value was calculated using a log-rank test.

I  Schematic diagram of generating murine SCC tumour cells from the classic KP (lsl-KrasG12D: p53 fl/fl) mouse model.

J  Representative qPCR of SCC and ADC marker expression in two independent KP lung tumour clones, resulting in KPADC and KPSCC (ACTIN served as loading control). Quantitative graph is represented as mean ± SD of three experiments ($n$ = 3).

K  Immunoblot against endogenous USP28 in the murine KPSCC cell line upon targeting with two independent shRNA sequences. ACTIN served as loading control.

L  Schematic diagram of orthotopic re-transplantation of cell lines into the lung of recipient wild-type C57BL/6J mice.

M  Representative haematoxylin and eosin (H&E) images of tumour-bearing animals 8 weeks post-intratracheal transplantation of $2 \times 10^{5}$ cells/animal. KPSCC sh-NTC ($n$ = 3); KPSCC sh-USP28 ($n$ = 3), scale bar = 5,000 μm.

N  Quantification of % tumour area (top, normalized to total lung area) and lung tumour numbers (bottom) on KPSCC sh-NTC ($n$ = 6) and KPSCC sh-USP28 ($n$ = 6) animals.

O  Representative IHC staining for ADC (TTF-1) and SCC (KRT5 and $\Delta$Np63) marker expression as well as Usp28 and GFP abundance in KPSCC sh-NTC ($n$ = 3) and KPSCC sh-USP28 ($n$ = 3) lung tumours. Scale bar = 50 μm.

Data information: In box plots, the centre line reflects the median, the cross represents the mean, and the upper and lower box limits indicate the first and third quartiles. Whiskers extend 1.5× the IQR. $P$-values were calculated using two-tailed $t$-test. **$P$ < 0.01; ****$P$ < 0.0001. See also Fig EV3 and Appendix Table S3 (exact $P$-values and statistical test used).

Source data are available online for this figure.

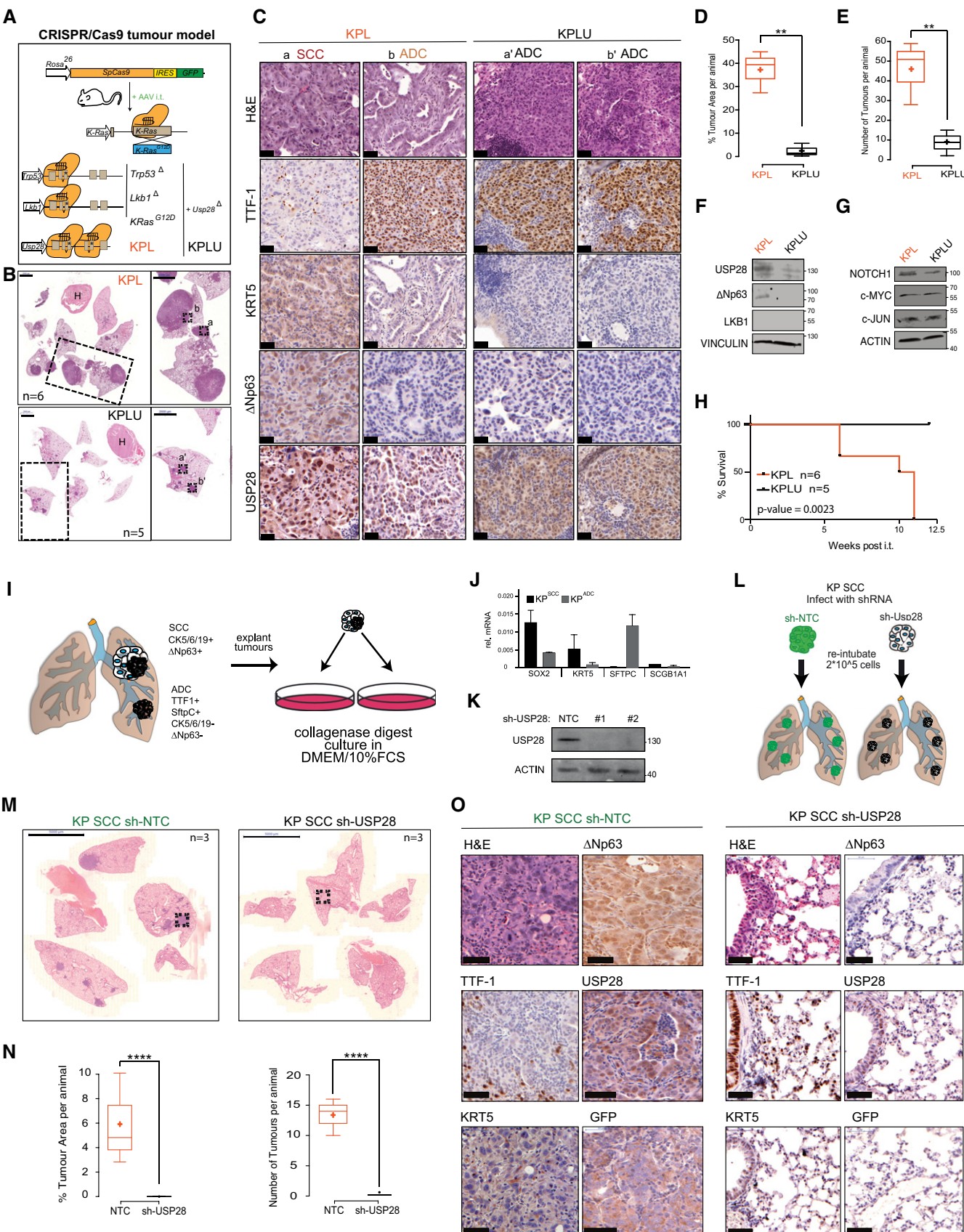

**Figure 5.**

showed clear signs of tumour formation (Fig EV3I). Notably, we observed both NSCLC tumour entities, ADC and SCC (Fig EV3J). Single lung tumour nodules were macroscopically excised from the affected lung lobes, collagenase-treated and cultured in standard tissue culture medium (Fig 5I). Adherent cells underwent several rounds of splitting and re-seeding, and established lines were tested for ADC and SCC marker expression (Fig 5J). One of the cell lines expressing SCC marker genes was then infected with lentiviruses expressing constitutive shRNAs targeting murine USP28. Western blot analysis of targeted cells demonstrated an almost complete ablation of USP28 protein in lentiviral shRNA-infected cells, when compared to controls (NTC; Fig 5K). 200,000 control or USP28 knock-down cells were then orthotopically implanted into the lungs of wild-type C57Bl/6J mice by tracheal intubation (Fig 5L). Tumour burden and histological type were analysed 8 weeks post-intratracheal transplantation. Animals transplanted with NTC cells developed tumours within 8 weeks (Fig 5M and N). These tumours resemble the marker expression of the primary cell line and maintained SCC lineage identity (TTF1$^-$/KRT5$^+$/ΔNp63$^+$). Moreover, expression of GFP, a marker for shRNA expression, was also readily detectable by IHC staining (Fig 5O). In contrast, mice transplanted with USP28-depleted cells failed to develop tumours, as only tumour-free lungs were observed (Fig 5M and N). The absence of cancerous cells was also confirmed by histological analysis (Fig 5O). Taken together, these data show that USP28 is required for lung SCC formation and engraftment.

### ΔNp63-driven SCC cells of various tissues are vulnerable to USP28 depletion

Chromosomal amplification and increased gene expression of ΔNp63 are very common in lung SCC when compared to cervix, head-and-neck, oesophagus or pancreatic tumours (Fig 6A). Nevertheless, ΔNp63 expression is increased in SCC irrespective of "tissue of origin" (Fig 6B and Appendix Fig S3A). Therefore, we were wondering if USP28 is also upregulated in these SCC. Publicly available expression datasets revealed an upregulation of *TP63* and *USP28* in cancer samples from cervix, oesophagus, head-and-neck

or lung SCC compared to non-transformed samples (Figs 1 and 6B). Patients with an increased expression of *USP28* showed a significantly shortened overall survival in cervix and head-and-neck tumours, while expression of ΔNp63 significantly shortened life expectancy in pancreatic cancer (Appendix Fig S3B). As USP28 is involved in the regulation of several proto-oncogenes, we wondered which factors regulate either SCC or ADC cell identity. Publicly available datasets revealed a positive correlation between KRT14 and TP63 in lung, cervix, oesophagus and pancreatic tumours (Fig 6C and Appendix Fig S3C). To address the specificity of gene expression, we analysed the correlation between KRT7, a marker for ADC, and TP63 in the same dataset. Here, negative correlation could be observed, stressing the specificity of ΔNp63 as an essential SCC transcription factor (Fig 6C). This observation was further validated by analysing the correlation of the USP28 targets with the consensus SCC marker gene signature (Fig 6D). We could identify that TP63 showed the strongest correlation with the SCC cell identity for all tumour subtypes analysed (Fig 6D). While USP28 regulates several proto-oncogenes in SCC, however, cells are susceptible to deregulated ΔNp63.

To investigate a therapeutic potential of targeting the USP28-ΔNp63 axis in SCC cells of different origins, we used a set of human cancer cell lines, comprising the pancreas lines PANC-1 (ADC) and BXPC-3 (SCC); cervical cancer cell lines HeLa (ADC), SiHa and Ca Ski (SCC); the head-and-neck cell line Detroit 562 (SCC); and the lung cell lines H1299 (ADC) and LUDLU-1$^{adh}$ (SCC). While USP28 was readily detectable by immunoblot in all cell lines (Appendix Fig S3D), only the SCC lines Ca Ski and BXPC3 and the head-and-neck cell line Detroit 562 expressed detectable amounts of endogenous ΔNp63. In the cervical line SiHa, despite being of SCC origin, endogenous ΔNp63 was not detectable (Appendix Fig S3D).

Next, we targeted USP28 by two independent shRNA sequences and analysed the knock-down efficacy by immunoblotting (Fig 6E and Appendix Fig S6E). Both shRNAs induced a significant decrease in USP28 protein levels in ADC and SCC of lung, cervix and pancreas (Fig 6E). While SCC cell lines infected with a non-targeting control shRNA expressed ΔNp63 and the downstream target KRT14, knock-down of USP28 reduced ΔNp63 protein levels, along with the

**Figure 6. ΔNp63-driven SCC cells of various tissues are vulnerable to USP28 depletion.**

A   Analysis of occurring TP63 genetic alterations in lung squamous (LUSC), cervical (CESC), oesophagus (ESCA), head-and-neck (HNSC) and pancreatic (PAAD) tumours. CBioPortal.

B   Expression of TP63 (left) and USP28 (right) in human lung (*n* = 498), cervix (*n* = 254), oesophagus (*n* = 96) and HNSC (*n* = 522) SCC tumours and normal non-transformed tissue (nLung = 338 nCervix = 3, nOesophagus = 11 and nHNSC = 44). In box plots, the centre line reflects the median, the cross represents the mean, and the upper and lower box limits indicate the first and third quartiles. Whiskers extend 1.5× the IQR. *P*-values were calculated using two-tailed *t*-test statistical analysis. Xena UCSC software.

C   Correlation of mRNA expression of KRT14/KRT7 and TP63 in ADC and SCC tumours for lung (nADC = 513 and nSCC = 498), cervix (nADC = 47 and nSCC = 254) and oesophagus (nADC = 89 and nSCC = 96). Blue dots: ADC; red dots: SCC; R = Spearman's correlation coefficient; m: slope. Xena UCSC.

D   Spearman's correlation values of the gene list "Squamous cancer markers" and TP63, c-JUN, c-MYC or NOTCH1 in lung (*n* = 1,011), cervix (*n* = 301), oesophagus (*n* = 185), HNSC (*n* = 522) and PAAD (*n* = 178) tumours. Intense blue: low correlation; intense red: strong correlation. GEPIA software.

E   Immunoblot of control (sh-NTC) and two independent shRNA targeting USP28 (sh-USP28#1 and #2) for ΔNp63, KRT14 and USP28 protein abundance in H1299, LUDLU-1$^{adh}$, HeLa, SiHa, Ca Ski, PANC-1 and BXPC-3 (ACTIN as loading control).

F   Cells were seeded at equal cell density and counted after 5 days, bright-field images of control or sh-USP28#2-infected H1299, LUDLU-1$^{adh}$, HeLa, Ca Ski, PANC-1 and BXPC-3 cells before quantification. Scale bar = 30 μm.

G   Relative number of H1299, LUDLU-1$^{adh}$, HeLa, Ca Ski, SiHa, PANC-1 and BXPC-3 sh-USP28#2 cells compared with sh-NTC control cells. *P*-values were calculated using two-tailed *t*-test statistical analysis. SiHa* = notably, the human SCC cell line SiHa was negative for ΔNp63.

Data information: All quantitative graphs are represented as mean ± SD of three experiments (*n* = 3). *P*-values were calculated using two-tailed *t*-test statistical analysis; *$P$ < 0.05, ****$P$ < 0.0001. See also Appendix Fig S3 and Appendix Table S3 (exact *P*-values and statistical test used).
Source data are available online for this figure.

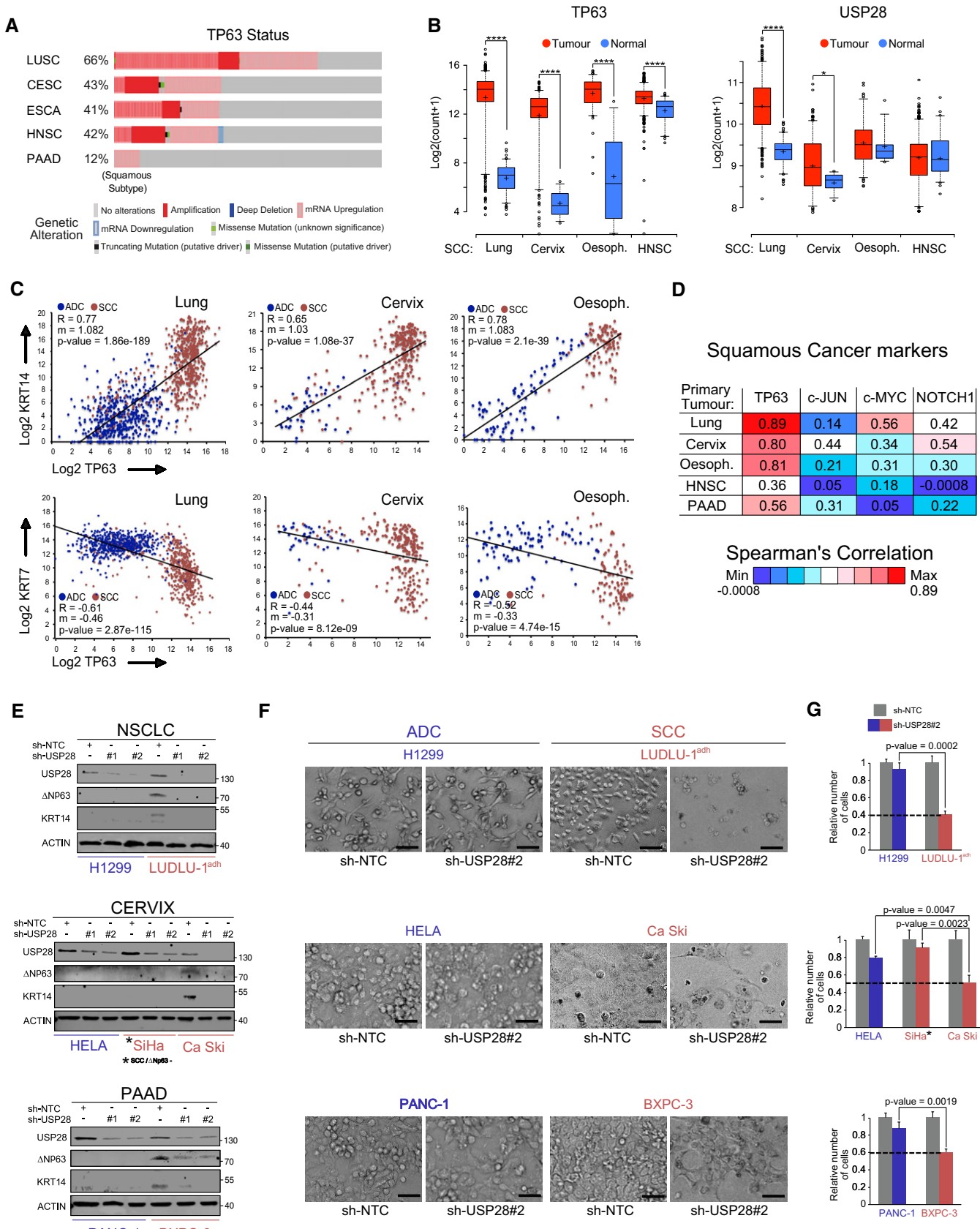

**Figure 6.**

SCC marker KRT14 (Fig 6E and Appendix Fig S3E), consistent with observed USP28 effects in A-431 cells (Fig EV2).

SCC is uniquely dependent on the expression of ΔNp63; therefore, we next analysed the effect of USP28 knock-down on ADC and SCC cell line proliferation (Fig 6F and G, Appendix Fig S3F and G). Control or USP28-depleted ADC and SCC cells were seeded at equal cell density and counted after 5 days. Proliferation of the ADC cells HeLa, PANC-1 and H1299 was only weakly affected (Fig 6F and G). Cells of SCC origin, which maintained expression of ΔNp63 (Ca Ski, BXPC3, Detroit 562 and LUDLU-1[adh]), demonstrated a growth disadvantage upon knock-down of USP28 by shRNA (Fig 6F and G, Appendix Fig S3F). An exception was the SCC cell line SiHa, which does not express ΔNp63, and only showed an anti-proliferative effect similar to cells of ADC lineage (Fig 6E and G, Appendix Fig S3G).

## Pharmacologic inhibition of USP28 regulates ΔNp63 stability via deubiquitylation and shows a selective anti-proliferative response of SCC cells

Recently, small-molecule inhibitors of ubiquitin-specific proteases, including USP28, have become available (Lamberto *et al*, 2017; Wrigley *et al*, 2017). Therefore, we tested the available inhibitor AZ1 to target USP28 activity and, subsequently, to deregulate ΔNp63 protein stability in SCC. Firstly, we established the required dosage of AZ1 cells to block USP28 activity in A-431. Cells were treated for 24 h with different concentrations of AZ1 and USP28 activity was assessed by ubiquitin-suicide probes (Warheads), followed by immunoblotting against USP28 (Fig 7A). In non-treated cells, the majority of USP28 was in an active state, however, AZ1 blocked USP28 activity in a dose-dependent fashion (Fig 7A). Next, we tested the effect of USP28 inhibition on ΔNp63 protein abundance. A-431 cells were cultured in the presence of AZ1 for 24 h and protein abundance was analysed by Western blot. In a dose-dependent fashion total USP28 abundance was reduced ($IC_{50} \sim 18.8$ μM) concomitant with a reduction in ΔNp63 ($IC_{50} \sim 14.6$ μM) and the ΔNp63 target KRT14 (Fig 7B). USP28 inhibition results in its degradation and treatment with AZ1 showed a significant decrease in USP28 at higher concentrations ($IC_{50} > 18$ μM). This is in line with previous observations that the enzymatic activity of DUBs is required to enhance their own stability (de Bie & Ciechanover, 2011; Wang *et al*, 2017).

Next, we wondered if the ubiquitylation of ΔNp63 was affected by AZ1 (Fig 7C). A-431 cells were treated with different concentrations of AZ1 for 24 h and endogenous ubiquitylation of ΔNp63 assessed by TUBE ubiquitin pulldown, followed by immunoblotting against ΔNp63. Addition of AZ1 enhanced the ubiquitylation of ΔNp63 in A-431 cells concentration-dependent (Fig 7C), indicating that the inhibitor AZ1 blocks the ability of USP28 to stabilize ΔNp63 by deubiquitylation. Since we identified USP28 interacting with several lysine-containing domains within ΔNp63 (Appendix Fig S1B), we were wondering if the observed effects of AZ1 on ΔNp63 stability depend on these sites. Therefore, we designed constructs, in which all 21 lysine residues were either mutated to arginine (ΔNp63[KtoR]) or depleted (ΔNp63[Kless]; Fig EV4A). While FLAG-tagged ΔNp63 was sensitive to USP28 inhibition via AZ1 treatment, neither FLAG-ΔNp63[KtoR] nor ΔNp63[Kless] showed decrease in protein stability upon treatment with AZ1 (Fig EV4B). These data suggest

that the observed effects of AZ1 on ΔNp63 depend on the presence of ubiquitin acceptor sites and the degradative effect is mediated by inhibition of USP28 in a AZ1-dependent fashion.

Next, we tested if exposure to AZ1 induces the degradation of ΔNp63 via the 26S proteasome. A-431 cells were pre-treated for 18 h with AZ1 and either EtOH or MG132 was added for further 6 h prior to cell collection and subsequent Western blot. In the presence of AZ1, USP28 as well as ΔNp63 were degraded (Fig 7D). Upon addition of MG132, USP28 and ΔNp63 protein levels were restored, but lower than in control cells (Fig 7D). This observation indicates that the treatment with AZ1 induces both the inhibition and proteasomal degradation of USP28, thereby mediating the destabilization and loss of ΔNp63.

As SCC cells require ΔNp63 to maintain proliferation (Abraham *et al*, 2018) and genetic depletion of USP28 by shRNA affected proliferation of human SCC cells (Fig 6), we investigated the ability of AZ1 to hinder growth in A-431. Twenty-four hours post-seeding, cells were grown in the presence of increasing concentrations of AZ1 for an additional 48 h, and cell numbers were measured by Hoechst immunofluorescence (Fig 7E). While A-431 tolerated low concentrations of AZ1, in a dose-dependent fashion, however, cells stopped to proliferate ($GI_{50} \sim 18$ μM).

The ability of AZ1 to destabilize and thereby deregulate ΔNp63 protein abundance was further validated in the SCC cell lines LUDLU-1[adh], Ca Ski, Detroit 562 and BXPC-3 (Figs 7F and EV4C). When exposed for 24 h to increasing concentrations of AZ1, USP28, ΔNp63 as well as the SCC marker KRT14 were downregulated in all tested SCC cells (Figs 7F and EV4C). Finally, we targeted the USP28-ΔNp63 axis via AZ1 in SCC cell lines and compared proliferative responses to a respective ADC line (Figs 7G and H, and EV4D). ADC and SCC cells were seeded at equal cell density, exposed to AZ1 for 48 h and counted. Proliferation of the ADC cells H1299, HeLa or the SCC line SiHa was affected at higher concentrations (Fig 7G and H). ΔNp63 expressing SCC cells LUDLU-1[adh] and Ca Ski, as well as Detroit 562 and BXPC3, demonstrated a growth disadvantage at significantly lower concentrations (Figs 7G and H, and EV4E and F).

AZ1 selectively sensitizes SCC cells, in particular when compared to ADC, of various origins. These data demonstrate that AZ1 blocks USP28-dependent stabilization of ΔNp63 and induces its proteasomal degradation.

## Pharmacologic inhibition of USP28 with AZ1 reduces tumour growth in an orthotopic model of lung SCC tumours

Since ΔNp63 expressing SCC cells showed a selective, anti-proliferative response to AZ1 treatment, we wondered if *in vivo* tumour maintenance could be affected. To address this question, we orthotopically transplanted murine lung tumour cells established from the previously used mouse model (described in Fig 5, KPL) in wild-type C57BL/6J mice. In contrast to lung tumour cells established from the KP CRISPR/Cas9 mouse model, KPL tumour cells expressed ΔNp63 (Fig 8A). Twenty-eight days post-transplantation, mice were grouped in three cohorts ($n = 3$ per group), control animals receiving solvent alone (PBS/DMSO/Tween-80), and two experimental groups receiving 125 mg/kg AZ1 or 375 mg/kg AZ1, respectively. Doses were given every 3 days via intraperitoneal injection (i.p.) for a total of 6 injections (Fig 8B). Animals were sacrificed and immunohistologically analysed. Mice tolerated the application of control

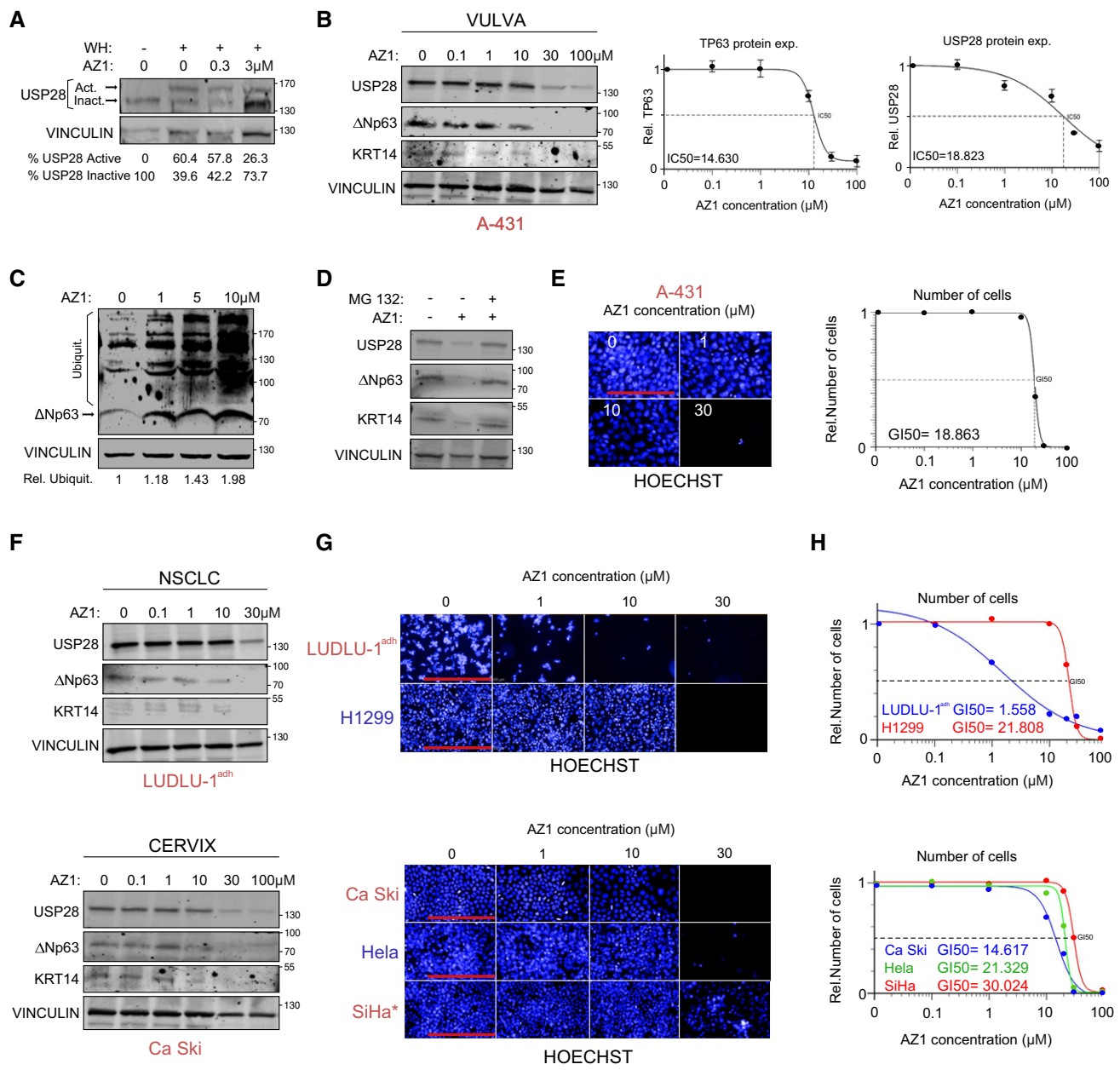

**Figure 7. Pharmacological inhibition of USP28 with the first-generation inhibitor AZ1 regulates ΔNp63 stability via deubiquitylation and shows a selective anti-proliferative response of SCC cells.**

A   Immunoblot of endogenous USP28 in A-431 cells upon treatment with warheads (WH) and either DMSO or indicated concentrations of AZ1 (USP28 Inh.) for 24 h. USP28 Upper band: active USP28; USP28 lower band: inactive USP28, VINCULIN as loading control.

B   Immunoblot of endogenous USP28, ΔNp63 and KRT14 in A-431 cells treated for 24 h with either DMSO or indicated concentrations of AZ1. VINCULIN served as loading control. TP63 and USP28 half-maximal inhibitory protein abundance (IC50) was calculated. Graphs are represented as mean ± SD (n = 3).

C   TUBE pulldown of endogenous ubiquitylated ΔNp63 in A-431 cells upon treatment with either DMSO or indicated concentrations of AZ1 for 24 h. VINCULIN served as loading control for relative Ubiquitination.

D   Immunoblot of endogenous USP28, ΔNp63 and KRT14 in A-431 cells treated for 18 h with either DMSO or 15 μM AZ1 followed by 6 h of treatment with 20 μM proteasomal inhibitor MG-132 (20 μM) or DMSO. VINCULIN served as loading control.

E   A-431 cells were seeded at equal cell density and cultured in the presence of either DMSO, 1 μM, 10 μM or 30 μM AZ1 for 48 h. Cells were quantified with the Operetta imaging system using Hoechst staining. 50% growth inhibition (GI50) was calculated. Scale bar = 250 μm. n = 30 fields analysed.

F   Immunoblot of endogenous USP28, ΔNp63 and KRT14 in LUDLU-1[adh], H1299, Ca Ski, HeLa and SiHa cells treated for 24 h with either DMSO or indicated concentrations of AZ1. VINCULIN served as loading control.

G   LUDLU-1ADH, H1299, Ca Ski, HeLa and SiHa cells were seeded at equal cell density and cultured in the presence of either DMSO, 1, 10 or 30 μM AZ1 for 48 h. Scale bar for LUDLU-1[adh] and H1299 = 500 μm. Scale bar for Ca Ski, HeLa and SiHa = 250 μm. SiHa* = notably, the human SCC cell line SiHa was negative for ΔNp63.

H   Number of cells was quantified with the Operetta imaging system using Hoechst staining. 50% growth inhibition (GI50) was calculated. n = 30 fields analysed.

Data information: See also Fig EV4.

solvent, 125 mg/kg, or 375 mg/kg AZ1 via i.p. well and showed no macroscopic abnormalities upon sacrifice (Fig EV5A). Also, overall tissue architecture was not affected in treated animals. This is in line with previous reports using genetic whole-body acute depletion models of USP28 (Diefenbacher et al, 2014). As tumour cells originated from constitutive Rosa[26Sor-CAGG-Cas9-IRES-GFP] mice, transplanted cells maintained the expression of GFP (Fig 8C; arrows indicate individual tumour lesions). While control solvent-treated mice showed large tumour lesions at endpoint, treatment with AZ1 for the indicated time points resulted in a reduced tumour burden, in a dose-dependent manner (Fig 8D and E). As USP28 was detectable by immunohistochemistry in all tumours (Fig 8E), we assessed the effect of AZ1 on USP28 activity. Small tumour lesions were excised from treated animals and the DUB activity assessed using ubiquitin-suicide probes, followed by immunoblotting against USP28 (Fig EV5B and C). In non-treated mice, active USP28 was detectable in *in vivo* tumour samples, however, treatment of mice with AZ1 blocked USP28 activity, and resulted in a marked reduction in USP28 protein abundance (Fig EV5C). As a consequence, while tumours of control-treated mice showed a strong staining against ΔNp63, mice treated with AZ1 showed a reduction in ΔNp63 abundance (Fig 8E and F). This was further confirmed by immunoblotting against ΔNp63 in isolated primary tumour lesions from control and AZ1-treated animals (Fig EV5C).

We concluded that pharmacologic targeting of USP28 in SCC via AZ1 is feasible and enables the targeting of ΔNp63, the major transcription factor and cell fate regulator in SCC, arguing that this axis establishes a unique dependence and hence may open a wide therapeutic window for targeting SCC of various tissue origins via USP28 (Fig 8G).

## Discussion

SCC tumours are among the genetically most complex entities (Cancer Genome Atlas Research N, 2012). Driver mutations can vary widely, ranging from activating mutations in members of the MAPK pathway, the PI3K pathway or RTKs, to gene amplifications in several loci, including ΔNp63 (Cancer Genome Atlas Research N, 2012). SCC tumours have in common an inherent dependence on ΔNp63 expression (Rocco et al, 2006; Bergholz & Xiao, 2012). Previous work has established the role of ΔNp63 as a master

transcription factor and regulator of SCC identity and proliferation (Abraham et al, 2018; Hamdan & Johnsen, 2018; Somerville et al, 2018). SCC tumour cells are addicted to ΔNp63 expression (Vivanco, 2014; Somerville et al, 2018), as depletion of ΔNp63 is not tolerated by these tumours and leads to rapid tumour regression (Ramsey et al, 2013). Therefore, targeting ΔNp63, either directly or by altering its protein abundance, appears promising for therapy of SCC tumours.

Most transcription factors, including ΔNp63 (Dang et al, 2017; Abraham et al, 2018; Lambert et al, 2018), are considered as "non-druggable" targets as their structure does not provide suitable domains for small-molecule interactions. Modulation of their abundance by targeting mechanisms that control protein stability presents a viable option (Liu et al, 2015; Wang et al, 2018). ΔNp63 is tightly regulated by the ubiquitin–proteasome system and ubiquitylated by various E3 ligases (Armstrong et al, 2016). In a tumour, this mechanism is frequently non-functional, leading to the accumulation of ΔNp63 protein (Ruiz et al, 2019) and results of the current study.

In this study, we explored the possibility to control the abundance of ΔNp63 protein via the deubiquitylase USP28. We found that USP28 is frequently upregulated in human SCC tumours and is often co-expressed with ΔNp63. USP28 regulates the abundance of ΔNp63 by direct binding and by catalysing the removal of K48-linked ubiquitin chains from the ΔNp63 protein. Loss of USP28 reduced SCC cell proliferation in a ΔNp63-dependent manner and interfered with the expression of marker genes that define the SCC lineage to a similar extent as targeting ΔNp63 directly. The observed effects in gene expression in the USP28 shRNA cell lines were a direct consequence of reduced ΔNp63 protein abundance as expression of exogenous ΔNp63 was able to restore SCC marker expression and cell proliferation in the USP28-targeted cells.

It was previously reported that USP28 is involved in the regulation of TP53 protein abundance in a USP28-TP53BP1 cell cycle-dependent fashion (Fong et al, 2016; Lambrus et al, 2016; Meitinger et al, 2016). *In vivo*, however, using a whole-body acute depletion model, in control and intestinal tumour-bearing animals, loss of *Usp28* did not result in changes of endogenous Tp53 nor in its activation (Diefenbacher et al, 2014). This observation argues that the observed effects upon interference with USP28 on cell proliferation in SCC are independent of TP53. This is in particularly relevant as

---

**Figure 8. Pharmacologic inhibition of USP28 with AZ1 reduces tumour growth in an orthotopic model of lung SCC tumours.**

A  Immunofluorescence (IF) staining against endogenous ΔNp63 in KP and KPL cell lines (blue: DAPI [staining of nuclei] and red/magenta: ΔNp63), scale bar = 32 μm.

B  Schematic diagram of AZ1 treatment. After first 28 days without treatment, tumours were treated with an AZ1 dosage every 3 days until day 46.

C  GFP Fluorescent images of lungs treated with either PBS/DMSO/0.1% Tween-80 or indicated concentrations of AZ1. Tumour cells are positive for GFP (green arrows). n = 3.

D  Quantification of % tumour area (normalized to total lung area) in control (n = 5), 125 mg/kg AZ1 (n = 4) and 375 mg/kg AZ1 (n = 4)-treated animals.

E  Representative staining of H&E and IHC for USP28 and ΔNp63 of SCC tumours from mice treated with PBS/DMSO/0.1% Tween-80 or indicated concentrations of AZ1. Boxes indicate highlighted tumour areas. First-line scale bars = 500 μm; lower scale bars = 20 μm. n = 3.

F  ΔNp63 staining intensity of lung tumours treated with PBS/DMSO/0.1% Tween-80 or indicated concentrations of AZ1. nControl = 33,712 cells; nAZ1 125 mg/kg = 7,382 cells; nAZ1 375 mg/kg = 4,967 cells. Cells were analysed from three independent animals.

G  Schematic model.

Data information: In box plots, the centre line reflects the median, the cross represents the mean, and the upper and lower box limits indicate the first and third quartiles. Whiskers extend 1.5× the IQR. P-values were calculated using two-tailed t-test. *P < 0.05; ****P < 0.0001. See also Fig EV5 and Appendix Table S3 (exact P-values and statistical test used).

Source data are available online for this figure.

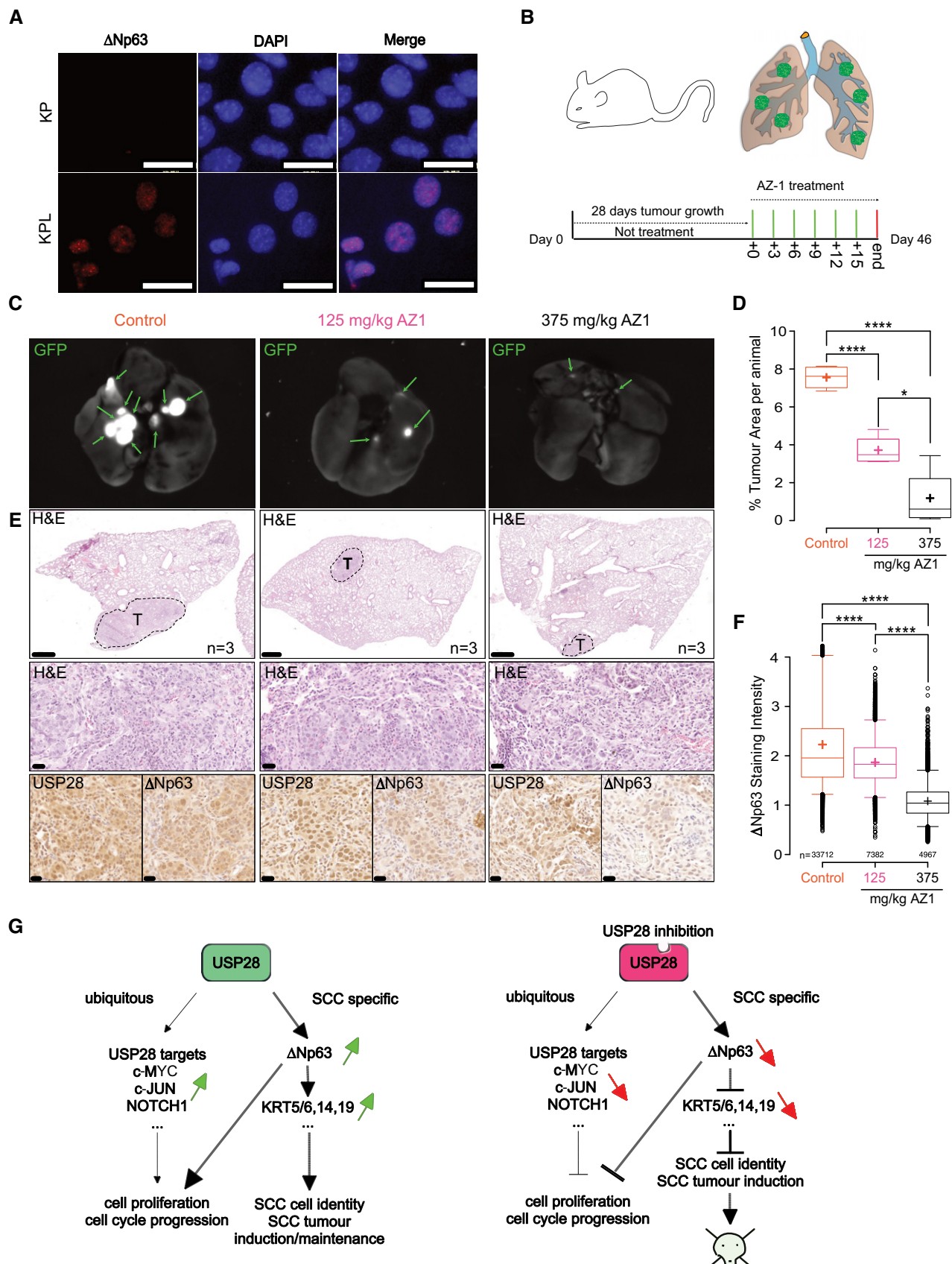

**Figure 8.**

the majority of patients diagnosed with lung squamous cell carcinoma harbour inactivating mutations within *TP53*.

The depletion of USP28 partially phenocopied the effect of ΔNp63 ablation. USP28 and ΔNp63 share a commonly regulated gene signature specific for SCC. USP28, however, regulates additional proto-oncogenes, and we detected changes in NOTCH signalling in USP28 shRNA cells, while JUN/AP-1 and MYC were not affected. Furthermore, genetic or pharmacologic targeting of USP28 recapitulated the genetic depletion of the master SCC transcription factor ΔNp63 and showed a high overlap with regard to essential biological processes essential to SCC tumours. Therefore, targeting USP28 is a good surrogate target for affecting ΔNp63 function in SCC.

In strong support of this notion, targeting USP28 in a model of lung SCC ("KPL mice") using the CRISPR/Cas9 system reduced the total number of tumours and, specifically, abrogated the formation of the SCC subtype. While KPL mice developed both NSCLC entities, ADC and SCC, loss of USP28 strongly affected overall tumour induction and blocked SCC formation. This indicates that SCC is strictly dependent on USP28 expression, consistent with their dependence on ΔNp63 function. USP28 was not only required for tumour induction, but also crucial for tumour maintenance, since upon USP28 depletion established SCC tumour cells completely failed to establish tumours in a syngeneic orthotopic transplantation experiment.

While the interaction between USP28 and ΔNp63 is independent of the catalytic site within the DUB, our results show that the active cysteine at position 171 is essential to facilitate the stabilizing effect on the ΔNp63 protein. Mutation of this cysteine residue strongly reduced the ability of USP28 to deubiquitylate ΔNp63. Since USP28 is known to homo-dimerize (Gersch *et al*, 2019; Sauer *et al*, 2019), overexpression of the catalytic dead isoform may dimerize with endogenous USP28, thereby functioning as a dominant negative mutant. Therefore, inhibiting the catalytic activity of USP28 is likely to be a suitable mechanism to target ΔNp63 in SCC tumours. That the DUB family presents a viable therapeutic option was tested by using a pan-DUB inhibitor, PR-619, as well as the USP25/USP28 dual-specific inhibitor AZ1 (Wrigley *et al*, 2017), in cell culture as well as *in vivo*. SCC cells exposed to the pan-inhibitor PR-619 showed a strong decrease in ΔNp63 levels. Exposure to AZ1 resulted in the inactivation of USP28 and degradation of endogenous ΔNp63 in a panel of human SCC lines, leading to growth arrest at low concentrations, while ADC cell lines, despite showing inactivation of USP28, tolerated higher concentrations. This was further confirmed by observing the ability of AZ1 to reduce tumour burden in a ΔNp63-expressing transplant lung cancer model system *in vivo*. While AZ1 had little effect to the overall wellbeing and macroscopic tissue architecture of treated animals, which is in line with previous reports of the tolerance of acute genetic whole-body depletion of USP28, tumour cells depended on the ability to maintain USP28 protein levels high and active, resulting in smaller tumours and reduced ΔNp63 protein levels at endpoint.

In addition to ΔNp63, targeting of USP28 reduced NOTCH signature gene expression and NOTCH1 protein abundance, consistent with previous data (Popov *et al*, 2007a; Diefenbacher *et al*, 2014, 2015; Schulein-Volk *et al*, 2014). As NOTCH signalling plays an important role in KRas-induced tumour formation in the lung (Xu *et al*, 2014b; Sosa Iglesias *et al*, 2018), loss of USP28 reduced

overall tumour burden and, while blocking SCC formation, also reduced ADC tumour burden in the lung. Similar effects were reported for genetic inhibition of NOTCH via overexpression of a dominant negative Mastermind-like protein (Xu *et al*, 2014b). This observation raises the possibility of a cooperative axis between ΔNp63 and NOTCH signalling via USP28 in SCC.

The transcriptional responses in SCC cells, however, are dominated by its effects on ΔNp63, arguing that this axis establishes a unique dependence and hence may open a wide therapeutic window for targeting SCC of various tissue origins via USP28. Given the crucial dependency of SCC tumour from different tissues, including those of head and neck, cervix, oesophagus, vulva and lung, on ΔNp63 to maintain proliferation and SCC cell identity, targeting the dependence of ΔNp63 via USP28 is a promising strategy to treat this tumour entity independent of tissue origin.

## Materials and Methods

### Tissue culture, transfection, infection and reagents

Cells were plated on Greiner Petri dishes and maintained at 37°C, 95% relative humidity and 5% $CO_2$ for optimal growth conditions. All cell lines were obtained from ATCC or ECACC. A-431, PANC-1, SiHa, Ca Ski, Detroit 562 and HEK-293T cells were cultured in DMEM (Gibco) supplemented with 10% foetal bovine serum (FBS) and 1% Pen-Strep. LUDLU-1[adh], NCI-H1299 and BxPC-3 cells were cultured in RPMI 1640 (Gibco) supplemented with 10% FBS, 1% GlutaMAX and 1% Pen-Strep. All the cells were passaged seven times as maximum to maintain cell identity. All cell lines were authenticated by STR profiling (conducted November 2019). Cells were tested for mycoplasma infection by PCR and were found to be not infected.

The human lung cancer cell line LUDLU-1 is maintained as a semi-attached/floating culture. We subjected this cell line to a selection process, enriching for an adherent clone, which was further propagated and used for all experiments within this study. Therefore, to highlight the difference towards the parental cell line, we decided to mark this by adding the suffix adh., hence naming the cell line LUDLU-1[adh]. The selection process did not alter the expression of endogenous ΔNp63 and SCC markers. It is noteworthy that the overall STR profile of the created subclone is similar, but not identical, to the parental cell line (Please see STR profiles in supplementary data).

For DNA transfection, a mix of 2.5 μg plasmid DNA, 200 μl free medium and 5 μl PEI was added into the 6-well dish medium (60% confluence); after 6-h incubation at 37°C, the medium was changed to full supplemented medium. For DNA infection, AAVs or lentiviruses (MOI = 10) were added to the cell medium in the presence of polybrene (5 μg/ml) and incubating at 37°C for 96 h. After incubation, infected cells were selected with 2.5 μg/ml puromycin for 72 h, 250 μg/ml neomycin for 2 weeks or FACS-sorted for RFP/GFP-positive cells (FACSCanto II BD).

Except when a different concentration was expressly indicated, the next reagents were dissolved in dimethyl sulfoxide (DMSO) or 70% ethanol and added to the cells at the following concentrations: cycloheximide (CHX; 100 μg/ml), doxycycline (DOX; 1 μg/ml) and tandem ubiquitin binding entity (TUBE; 100 μg/ml).

**Primary murine lung cancer cell lines**

In brief, at endpoint of experiment, tumour-bearing mice were sacrificed and lung lobes excised. The tissue was briefly rinsed in PBS and transferred to PBS-containing Petri dishes. By using a binocular, macroscopically detectable tumour lesions on the lung lobes were excised with a scissor and transferred to a test tube containing collagenase I (100 U/ml in PBS). The tumour-containing tissue was digested for 30 min at 37°C, and the reaction was stopped by addition of 10 l% FBS. The tissue/Collagenase/FBS mixture was briefly spun in a benchtop centrifuge and the supernatant discarded. Digested tissue was re-suspended in 10% FBS (Sigma), DMEM (Gibco/Thermo Fisher) and Pen/Strep (Sigma) and washed three times in 1 ml solution prior to plating in a 6-well tissue culture plate. During subsequent re-plating, fibroblasts were counterselected, by selective trypsinization, and cell clusters with a homogenous morphology were clonally expanded. These clones were then subjected to further biochemical analysis and characterization, including genotyping PCR, RNA sequencing and mass spectrometry. According to marker expression, cells were categorized as adenocarcinoma (ADC), squamous cell carcinoma (SCC) or adenosquamous carcinoma (ADSCC).

**RT–PCR**

RNA was isolated with peqGOLD TriFast (Peqlab), following the manufacturer's instructions. RNA was reverse-transcribed into complementary DNA (cDNA) using random hexanucleotide primers and M-MLV reverse transcriptase (Promega). Quantitative RT–PCR was performed using QPCR SYBR Green Mix (ABgene) on the instrument "Step One Realtime Cycler" (ABgene). RT–PCR was performed using the following program: 95°C for 15 min, 40× (95°C for 15 s, 60°C for 20 s and 72°C for 15 s), 95°C for 15 s and 60°C for 60 s. Relative expression was generally calculated with $\Delta\Delta C_t$ relative quantification method using the expression of B-ACTIN as housekeeping gene. For mouse USP28 mRNA expression in A-431, relative expression was calculated using $\Delta C_t$. For all the experiments, melt curve was performed. Primers used for this publication are listed.

**Immunological methods**

Cells were lysed in RIPA lysis buffer (20 mM Tris–HCl pH 7.5, 150 mM NaCl, 1 mM Na2EDTA, 1 mM EGTA, 1% NP-40 and 1% sodium deoxycholate), containing proteinase inhibitor (1/100) by sonication using Branson Sonifier 250 with a duty cycle at 20%, output control set on level 2 and the timer set to 1 min (10 sonication cycles per sample). Protein concentration was quantified using Bradford assay. After mixing 1 ml of Bradford reagent with 1 μl of sample, the photometer was used to normalize the protein amounts with a previously performed bovine serum albumin (BSA) standard curve. The quantified protein (40 μg) was boiled in 5× Laemmli buffer (312.5 mM Tris–HCl pH 6.8, 500 mM DTT, 0.0001% Bromphenol blue, 10% SDS and 50% Glycerol) for 5 min and separated on 10% Tris gels in running buffer (1.25 M Tris base, 1.25 M glycine and 1% SDS). After separation, protein was transferred to polyvinylidene difluoride membranes (Immobilon-FL) in transfer buffer (25 mM Tris base, 192 mM glycine and 20% methanol) and

then incubated with blocking buffer (0.1% casein, 0.2×PBS and 0.1% Tween-20) for 45 min at RT. After Blocking, membranes were incubated with indicated primary Abs (1/1,000 dilution in a buffer composed by 0.1% casein, 0.2× PBS and 0.1% Tween-20) for 4H at room temperature (RT). Secondary Abs (1/10,000 dilution in a buffer composed by 0.1% casein, 0.2× PBS, 0.1% Tween-20 and 0.01% SDS) were incubated for 1H at RT. Membranes were recorded with Odyssey® CLx Imaging System or iBright™ FL1000 Imaging System. Analysis and quantifications of protein expression were performed using Image Studio software (LI-COR Sciences). Data are presented as mean and error bars as standard deviation of the biological replicates. Significance was calculated with Excel using heteroscedastic two-tailed Student's $t$-tests. Immunoprecipitation was performed using 0.25 mg of Pierce™ Protein A/G Magnetic Beads (Thermo Fisher), 1 μg of the specific Ab and 500 μg of protein lysate. For endogenous co-immunoprecipitations, beads were incubated with Rabbit IgG as a control for specificity. For TUBE assay, 100 μg/ml of recombinant-expressed GST-4× UIM-ubiquilin fusion protein was mixed with the cell lysate before protein extraction and then processed for Western blot detection. Antibodies and dilutions used for this publication are listed (Appendix Tables S4 and S5).

**SPOT synthesis and interaction assay**

SPOT peptide membranes (Frank, 1992) were synthesized on an automated high-throughput parallel peptide synthesizer (MultiPep RSi, Intavis AG). The robot was loaded with acid-resistant 9-fluorenylmethyloxycarbonyl-$\beta$-alanine (Fmoc-$\beta$-Ala) functionalized 8 × 12 cm membranes (Intavis AG). Synthesis was initiated by Fmoc deprotection by applying two times 100 μl 20% piperidine in N-methylpyrrolidone (NMP) for 5 min followed by washing with dimethylformamide (DMF, 8 × 1.5 ml) and ethanol (EtOH, 6 × 1.5 ml and 2 × 2.4 ml) per 8 × 12 cm membrane. Peptide chain elongation was achieved using 100 nl of coupling solution consisting of pre-activated amino acids (0.5 M) with ethyl 2-cyano-2-(hydroxyimino)acetate (Oxyma, 1.5 M) and N,N′-diisopropylcarbodiimide (DIC, 1.1 M) in NMP (2:1:1, aa:Oxyma:DIC). The couplings were carried out four times for 12 min, respectively. Subsequently, the membrane was capped twice with 100 μl capping mixture (5% acidic anhydride in NMP) followed by washing with DMF (8 × 1.5 ml) and ethanol (EtOH 2 × 1.5 ml and 2 × 2.4 ml) per membrane. After chain elongation, final Fmoc deprotection was achieved by applying three times 20% piperidine in NMP (100 μl, 5 min each), followed by washes with DMF, 10 × 1.5 ml, EtOH, 8 × 1.5 ml and 2 × 2.4 ml. After vacuum-drying, SPOT membranes were removed from the robot and treated with 40 ml side-chain deprotection solution consisting of 80% TFA, 12% DCM, 5% H₂O and 3% TIPS for 1.5 h at room temperature. After washing with 4 × 20 ml DCM, 3 × 20 ml DMF and 3 × 20 ml EtOH and 1 × 20 ml 1 M acetic acid and 3 × 20 ml EtOH for 5 min, respectively, the SPOT membranes were stored at −20°C. Prior to whole protein lysate incubation, SPOT membranes were blocked using 5% bovine serum albumin/TBS for 1 h at room temperature. In parallel, A-431 cells were lysed using RIPA buffer (see section above). Upon blocking, the SPOT membrane was incubated with 1 mg total lysate dissolved in 5% BSA/TBS and incubated overnight at 4C shaking. Next day, membranes were

washed three times with washing buffer (0.2× PBS, 0.1% Tween-20), followed by incubation with indicated primary Abs (1/1,000 dilution in a buffer composed by 0.1% casein, 0.2× PBS and 0.1% Tween-20) for 4H at room temperature (RT). Secondary Abs (1/10,000 dilution in a buffer composed by 0.1% casein, 0.2× PBS, 0.1% Tween-20 and 0.01% SDS) were incubated for 1H at RT. Membranes were recorded with Odyssey® CLx Imaging System or iBright™ FL1000 Imaging System.

## Nickel-NTA-agarose his-ubiquitin purification

Transfected cells were lysed in 1 ml of buffer A (6M guanidine HCl, 0.1 M $Na_2HPO_4$, 0.1 M $NaH_2PO_4$, 0.01 M Tris (pH 8.0), 10 mM β-mercaptoethanol) per 100-mm dish 24 h after removal of the precipitate. The lysate was sonicated for 30 s to reduce viscosity and then mixed on a rotator with 50 µl (settled volume) of nickel-NTA-agarose (Qiagen, Chatsworth, CA) for 3 h at room temperature. The beads were washed three times with 1 ml of buffer A, twice with 1 ml of buffer A diluted in 25 mM with Tris–HCl (pH 6.8)/20 mM imidazole 1:4 and twice with 1 ml of 25 mM Tris–HCl (pH 6.8)/20 mM imidazole. Purified proteins were eluted by boiling the beads in 2× sample buffer supplemented with 200 mM imidazole and analysed by immunoblotting.

## Purification of endogenous polyUb conjugates by TUBE pulldown

The TUBE affinity reagent was expressed in *E. coli* as previously described (Damgaard *et al*, 2019). Endogenous polyUb conjugates were purified from cells/tissues using TUBE affinity reagents as described previously (Fiil *et al*, 2013). Briefly, cells/tissues were lysed in buffer containing 20 mM Na2HPO4, 20 mM NaH2PO4, 1% (v/v) NP-40 and 2 mM EDTA and supplemented with 5 mM NEM, and complete protease inhibitor cocktail (Bimake). TUBE (50 µg/ml) was added directly to the lysis buffer immediately before lysis, and cells were scraped off the plates, mixed by pipetting five times and lysed on ice for 30 min. Lysate was cleared by centrifugation, glutathione Sepharose 4B beads (GE Healthcare) were added, and pulldown was performed for 4–16 h at 4°C on rotation. The beads were then washed four times in 1 ml of ice-cold PBS + 0.1% (v/v) Tween-20, and bound material was eluted by mixing the beads with 1× sample buffer and heating to 50°C for 5 min.

## Immunofluorescence

Standard procedures were used for immunohistochemistry (IHC) and immunofluorescence (IF). For antibodies used in IHC or IF, manufacturer's manuals and instructions were used regarding concentration or buffer solutions. All primary antibodies were incubated overnight at 4°C, followed by washing thrice with PBS and subsequent incubation with the secondary antibody for 1 h at room temperature. IHC slides were imaged using Pannoramic DESK scanner and analysed using CaseViewer software (3DHIS-TECH). For samples stained with IF, tissue samples/cells were counterstained with 5 µg/ml DAPI, to highlight nuclei, for 15 min after secondary antibody application. Stained samples were mounted with Mowiol® 40–88 and imaged using a FSX100 microscopy system (Olympus).

## FACS, growth curves, Operetta system and IC50/GI50 calculation

FACS analysis and quantification were performed with FACSCanto II (BD) flow cytometer, and results were quantified and visualized using FlowJo software. For cell cycle analysis, cells were fixed using 70% ethanol and stained with propidium iodide (Sigma-Aldrich) at 4°C for 30 min. For cell number quantification, cell lines were seeded at 10% confluence in 24-well dishes. After 3.5, 5 or 7 days, tumour cells were trypsinized and counted with Invitrogen Countess II FL. Colony assays were performed by staining 70% ethanol-fixed cells with 0.5%. All the graphics except box plots were generated using Excel (Microsoft). Data were presented as mean and error bars as standard deviation of three independent biological replicates. For GI50, Operetta High-Content Imaging System was used; cells were seeded in 384-well dishes at equal density and treated with AZ1 at indicated concentrations for 48 h. Cells were fixed using 4% PFA for 10 min and then permeabilized using 0.5% Triton X-100 in PBS for 5 min. Before quantification, cells were stained with Hoechst. Number of cells was determined counting the number of nucleus. For the quantification, unhealthy cells with modified nuclear morphology were excluded. GI50 and IC50 were calculated and visualized using the website: www.aatbio.com.

## Site-directed mutagenesis sgRNA and shRNA design

sgRNAs were designed using the CRISPRtool (https://zlab.bio/guide-design-resources). shRNA sequences were designed using SPLASH algorithm (http://splashrna.mskcc.org/; Pelossof *et al*, 2017) or the RNAi Consortium/Broad Institute (www.broadinstitute.org/rnai-consortium/rnai-consortium-shrna-library). For site-directed mutagenesis, guidance and instructions from the kit GeneEditor™ *in vitro* Site-Directed Mutagenesis System (Promega) were followed.

## AAV and lentivirus production and purification

Virus was packaged and synthetized in HEK293-T cells seeded in 15-cm dishes.

For AAV production, cells (70% confluence) were transfected with the plasmid of interest (10 µg), pHelper (15 µg) and pAAV-DJ or pAAV-2/8 (10 µg) using PEI (70 µg). After 96 H, the cells and medium of three dishes were transferred to a 50-ml Falcon tube together with 5 ml chloroform. Then, the mixture was shaken at 37°C for 60 min and NaCl (1 M) was added to the mixture. After NaCl is dissolved, the tubes were centrifuged at 20,000× *g* at 4°C for 15 min and the chloroform layer was transferred to another Falcon tube together with 10% PEG8000. As soon as the PEG8000 is dissolved, the mixture was incubated at 4°C overnight and pelleted at 20,000× *g* at 4°C for 15 min. The pellet was re-suspended in PBS with $MgCl_2$ and 0.001% pluronic F68; then, the virus was purified using 1× chloroform and stored at −80°C. AAVs were titrated using Coomassie staining and RT–PCR using AAV-ITR sequence-specific primers.

For lentivirus production, HEK293 cells (70% confluence) were transfected with the plasmid of interest (15 µg), pPAX (10 µg) and pPMD2 (10 µg) using PEI (70 µg). After 96 H, the medium containing lentivirus was filtered (0.45 µM) and stored at −80°C.

### *In vivo* experiments and histology

All *in vivo* experiments were approved by the Regierung Unterfranken and the ethics committee under the licence numbers 2532-2-362, 2532-2-367 and 2532-2-374. The mouse strains used for this publication are listed. All animals are housed in standard cages in pathogen-free facilities on a 12-h light/dark cycle with *ad libitum* access to food and water. Animal health monitoring via sentinel animal screening is carried out in accordance with FELASA 2014 guidelines and conducted every 3 months.

Adult mice (around 8 weeks old) were anaesthetized with isoflurane and intratracheally intubated with 50 μl AAV ($3 \times 10^7$ PFU) or 200,000 cells diluted in PBS. Viruses were quantified using Coomassie staining protocol (Kohlbrenner *et al*, 2012), and cells were quantified with Invitrogen Countess II FL. For intratracheal instillation, a gauge 24 catheter was introduced to the trachea and cells or virus previously isolated was pipetted to the top of the catheter. During animal breathing, cells and virus were delivered into the lungs. For AZ1 treatment *in vivo*, previously indicated dosages at the indicated time points were inoculated intraperitoneally into the animals. AZ1 was dissolved in PBS/DMSO/0.1% Tween-80. At the indicated time point, animals were sacrificed by cervical dislocation and lungs were dissected and later fixed using 5% NBF. For AZ1-treated mice, lungs were scanned for GFP (iBright™ FL1000 Imaging System) before fixation. Samples were embedded in paraffin and sectioned at 6 μm using the microtome (Leica). Before staining, slides were de-paraffinized and rehydrated using the next protocol: $2 \times 5$ min in xylene, $2 \times 3$ min in EtOH (100%), $2 \times 3$ min in EtOH (95%), $2 \times 3$ min in EtOH (70%), 3 min in EtOH (50%) and 3 min in $H_2O$. Slides were stained with haematoxylin and eosin or immunohistochemistry (IHC) using the reported antibodies. For all staining variants, slides were mounted with 200 μl of Mowiol® 40-88 covered up by a coverslip on top. IHC slides were recorded using Pannoramic DESK scanner and analysed using CaseViewer software (3DHISTECH). IF samples were recorded using FSX100 microscopy system (Olympus). Quantification of the tumour area and number of tumours were performed using ImageJ as previously described (Jensen, 2013). Quantification of staining intensity for Fig 7 was performed using the software QuPath (https://qupath.github.io) (Bankhead *et al*, 2017) as indicated at the website.

### Human lung cancer samples

Lung cancer samples were obtained, stored and managed by Pathology Department Cordoba, Pathology Department University Hospital Würzburg and U.S. Biomax (lung microarray slides; slide LC2083). Informed consent was obtained from all subjects, and conducted experiments conform to the principles set out in the WMA Declaration of Helsinki and the Department of Health and Human Services Belmont Report. Samples provided from the hospitals in Cordoba and Würzburg are approved under ethical approval licences Decret 439/2010 (Hospital Universitario Reina Sofia) and Ethics approval 17/01/2006 (University Hospital Würzburg). Human tissue samples were stained against USP28 and ΔNP63. For quantification purposes, staining intensity was graded from 0 (no staining) up to 3 (staining intensity > 66%) by three independent pathologists. Box plots were generated using BoxPlotR online tool

(http://shiny.chemgrid.org/boxplotr/). The significance was calculated using *t*-test.

### RNA sequencing

RNA sequencing was performed with Illumina NextSeq 500 as described previously (Buchel *et al*, 2017). RNA was isolated using ReliaPrep™ RNA Cell Miniprep System Promega Kit, following the manufacturer's instruction manual. mRNA was purified with NEBNext® Poly(A) mRNA Magnetic Isolation Module (NEB) and the library was generated using the NEBNext® Ultra™ RNA Library Prep Kit for Illumina, following the manufacturer's instructions. For the size selection of the libraries, Agencourt AMPure XP Beads (Beckman Coulter) were used. Library quantification and size determination were performed using Fragment Analyzer (Agilent formerly Advanced Analytical).

### Quantification and statistical analysis

#### *RNA-sequencing analysis*
Fastq files were generated using Illumina's base-calling software GenerateFASTQ v1.1.0.64, and overall sequencing quality was analysed using the FastQC script. Reads were aligned to the human genome (hg19) using TopHat v2.1.1 (Kim *et al*, 2013) and Bowtie 2 v2.3.2 (Langmead & Salzberg, 2012), and samples were normalized to the number of mapped reads in the smallest sample. For differential gene expression analysis, reads per gene (Ensembl gene database) were counted with the "summarizeOverlaps" function from the R package "GenomicAlignments" using the "union" mode and non-expressed or weakly expressed genes were removed (mean read count over all samples < 1). Differentially expressed genes were called using edgeR (Robinson *et al*, 2010), and resulting *P*-values were corrected for multiple testing by false discovery rate (FDR) calculations.

#### *GSEA*
For gene set enrichment analyses (GSEA; Mootha *et al*, 2003; Subramanian *et al*, 2005), five gene sets were generated: "Genes Downregulated sh-USP28" ($q < 0.05$, $\log_2 FC < -1.5$), "Gene Up-regulated sh-USP28" ($q < 0.05$, $\log_2 FC > 1.5$), "Genes Down-regulated sh-ΔNP63" ($q < 0.05$, $\log_2 FC < -3$),"Genes Up-regulated sh-ΔNP63" ($q < 0.05$, $\log_2 FC > 3$) and "Squamous Cancer marker" (consensus list of upregulated genes for squamous tumours). Genes included in each gene set are reported in Appendix Tables S1 and S2. GSEAs were done with Signal2Noise metric and 1,000 permutations.

#### *Gene ontology analysis*
Gene ontology analysis was performed with PANTHER (Mi *et al*, 2019) using the "Statistical overrepresentation test" tool with default settings. Three gene sets were generated based on the RNA-seq data and analysed: "Genes Down-regulated sh-USP28" ($q < 0.05$, $\log_2 FC > 1.5$) and "Genes Down-regulated sh-ΔNP63" ($q < 0.05$, $\log_2 FC < -1.5$).

#### *Venn diagrams and pair-wise correlations*
Venn diagrams were visualized using the online tool: http://bioinformatics.psb.μgent.be/webtools/Venn/. *P*-values were calculated with a hypergeometric test using the online tool at http://nemates.org. Pearson's correlation was used for RNA-seq data from A-431.

*P*-values for Pearson's correlation were calculated using two-tailed Student's *t*-tests.

### Analysis of publicly available data

All publicly available data and software used for this publication are listed. (Appendix Table S5). Oncoprints were generated using cBioPortal (Cerami *et al*, 2012; Gao *et al*, 2013). Briefly, oncoprints generate graphical representations of genomic alterations, somatic mutations, copy number alterations and mRNA expression changes. The following studies were used for the different analysis: lung adenocarcinoma ("LUAD-TCGA, Provisional"), lung squamous cell carcinoma ("LUSC-TCGA, Provisional"), small-cell lung cancer ("U Cologne, Nature 2015"), lung normal samples (lung-GTEX), cervical cancer ("CESC-TCGA, Provisional"), oesophagus cancer ("ESCA-TCGA, Provisional"), head-and-neck tumours ("HNSC-TCGA, Provisional") and pancreatic cancer ("PAAD-TCGA, Provisional"). Data were obtained using UCSC Xena (preprint: Goldman *et al*, 2018). Data were downloaded as log2(norm_count+1).

Box plots using TCGA and GTEx data were generated using the online tool BoxPlotR (Kampstra, P. Beanplot: A Boxplot Alternative for Visual Comparison of Distributions. Journal of Statistical Software, Code Snippets 28(1). 1-9 (2008)) and GEPIA (Tang *et al*, 2017). For BoxPlotR, the data previously downloaded from UCSC Xena were used to generate the graphics, and *P*-values were calculated using two-tailed *t*-test. For GEPIA, the differential analysis was based on "TCGA tumours versus (TCGA normal + GTEx normal)", whereas the expression data were log2(TPM+1)-transformed and the log2FC was defined as median(tumour) – median(normal). *P*-values were calculated with a one-way ANOVA comparing tumour with normal tissue. Tukey and Altman whiskers were used depending on the number of samples.

Correlation analysis was calculated using GEPIA software or Excel with the data obtained from UCSC Xena. The analysis was based on the expression of the following datasets: "TCGA tumours", "TCGA normal" and "GTEx normal". The expression of USP28 and TP63 and the gene set "Squamous Cancer Markers" [consensus list of upregulated genes for squamous tumours based on previous publications (Wilkerson *et al*, 2010; Mukhopadhyay *et al*, 2014; Xu *et al*, 2014a; Ferone *et al*, 2016)] were used for the calculation of Spearman's correlation coefficients and significance by GEPIA software. *P*-values for correlation coefficients were calculated using two-tailed Student's *t*-tests.

The comparison of gene expression (using the dataset squamous cancer markers) across multiple tumour entities was done using GEPIA software based on the dataset "TCGA tumours". The colour code reflects the median expression of a gene in a tumour type, normalized with the maximum median expression across all different tumour types (row-wise *Z*-score).

Genomic signature comparing primary human lung tumour was performed using UCSC Xena (preprint: Goldman *et al*, 2018) based on the dataset "TCGA tumours".

Kaplan–Meier curves were estimated with the KM plotter (Nagy *et al*, 2018), cBioPortal (Gao *et al* 2013; Cerami *et al*, 2012) and R2: Genomics Analysis and Visualization Platform (http://r2.amc.nl). The KM plotter was used to analyse overall survival of lung cancer patients (Fig 1) based on gene expression data from microarrays obtained from GEO, caBIG and TCGA (REF). For R2: Genomics Analysis and Visualization Platform (http://r2.amc.nl), overall

**The paper explained**

**Problem**

Squamous cell carcinomas (SCCs) are among the genetically most complex and heterogeneous entities. While driver mutations can vary widely, all SCC have in common their intricate dependency on ΔNp63 expression. Previous work has unequivocally demonstrated that ΔNp63 is a master transcription factor that establishes SCC cell identity. In several SCC tumour models, it was demonstrated that tumours are addicted to ΔNp63 expression. Therefore, targeting ΔNp63, either directly or by altering its protein abundance, appears to be a promising strategy to tackle SCC tumours.

**Results**

In our study, we report that the deubiquitylase USP28 directly interacts and stabilizes ΔNp63 in SCC. Depletion of USP28 in human tumour cell lines affected proliferation and epithelial cell identity of SCC cells. This effect is mediated directly by destabilization of ΔNp63 protein upon loss of USP28. We document the dependence of SCC on USP28 using both cancer cell lines and *in vivo* murine lung tumour models. We determined that both proteins directly interact and that the enzymatic activity of USP28 is required to deubiquitylate, and stabilize, ΔNp63. *In vivo*, we could demonstrate that in a mouse model of lung SCC, loss of Usp28 during tumour initiation abolished SCC formation entirely. Similar effects could be demonstrated in an orthotopic lung SCC transplant model, where knock-down or inhibition of Usp28 was sufficient to hinder tumour growth. Finally, using pharmacologic inhibitors of USP28, we were able to specifically target SCC tumours by destabilizing ΔNp63.

**Impact**

Our work thus identifies for the first time a deubiquitylase, USP28, regulating ΔNp63 protein abundance. USP28 is druggable, and its modulation, *in vitro* and *in vivo*, negatively affected SCC in a ΔNp63-dependent manner; hence, USP28 can function as a druggable surrogate target for ΔNp63 in SCC. Our findings, by using currently available USP28 inhibitors, serve as a proof of concept that targeting SCC by targeting the DUB USP28 is a promising selective therapeutic strategy. If proved safe and efficient in humans, USP28 inhibitors could expand the current limited available portfolio of applicable SCC therapeutic agents.

survival and gene expression data were obtained from TCGA (Fig EV4). For the survival analysis of USP28-altered samples (mutation or deep deletion), cBioPortal was used to calculate disease-free survival using the dataset "Lung Squamous Cell Carcinoma (LUSC-TCGA, Provisional)" (Fig EV1). *P*-values were computed using a log-rank test.

## Data and software availability

RNA-sequencing data are available at the Gene Expression Omnibus under the accession number GEO (https://www.ncbi.nlm.nih.gov/geo/): GSE129982.

**Expanded View** for this article is available online.

## Acknowledgements

We are grateful to the animal facility and Barbara Bauer at the Biocenter, University Würzburg. C.P.G. and O.H. are supported by the German Cancer Aid

via grant 70112491, M.E. is supported by the TransOnc priority programme of the German Cancer Aid within grant 70112951 (ENABLE), and M.R. is funded by the DFG-GRK 2243 and IZKF B335. M.E.D. and A.O. are funded by the German-Israeli Foundation grant 1431. T. F. is funded by the IZKF programme Z2/CS-1.

## Author contributions

Conceptualization: CP-G, ME, and MED; methodology: CP-G (*in vitro*) and OH (*in vivo*); CP-G (Biochemistry); MRe (TUBE); FB (FACS); CS-V and UE (Operetta system); HMM (Peptide SPOT); CM (Mass Spec); and CPA (RNA-sequencing raw data analysis); formal analysis: SW and CP-G (Bioinformatics); MAC, MRo and MED (Pathology); and CM (Mass Spec); investigation: CP-G, OH, MRe, TF, FB, MRo and MED; resources: MAC, MRo and MED; writing—original draft: MED; writing—review and editing: CP-G, OH, MRe, AO, ME and MED; supervision: ME and MED; and funding acquisition: MED.

## Conflict of interest

The authors declare that they have no conflict of interest.

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
