## [Review Process File · EMBO Molecular Medicine]

Maintaining protein stability of Δ Np63 via USP28 is required by squamous cancer cells

Cristian Prieto-Garcia, Oliver Hartmann, Michaela Reissland, Fabian Braun, Thomas Fischer, Susanne Walz, Christina Schülein-Völk, Ursula Eilers, Carsten P. Ade, Marco A. Calzado, Amir Orian, Hans M. Maric, Christian Münch, Mathias Rosenfeldt, Martin Eilers & Markus E. Diefenbacher

Review timeline:	Submission date:	2nd Aug 2019
	Editorial Decision:	22nd Aug 2019
	Revision received:	19th Dec 2019
	Editorial Decision:	20th Jan 2020
	Revision received:	5th Feb 2020
	Accepted:	5th Feb 2020

Editor: Lise Roth

Transaction Report:

1st Editorial Decision

22nd Aug 2019

Thank you for the submission of your manuscript to EMBO Molecular Medicine. We have now received feedback from the three reviewers who agreed to evaluate your manuscript. As you will see from the reports below, the referees acknowledge the interest of the study and are overall supporting publication of your work pending appropriate revisions.

Addressing the reviewers' concerns in full will be necessary for further considering the manuscript in our journal, and acceptance of the manuscript will entail a second round of review. EMBO Molecular Medicine encourages a single round of revision only and therefore, acceptance or rejection of the manuscript will depend on the completeness of your responses included in the next, final version of the manuscript. For this reason, and to save you from any frustrations in the end, I would strongly advise against returning an incomplete revision.

When submitting your revised manuscript, please carefully review the instructions that follow below. Failure to include requested items will delay the evaluation of your revision:

- 1) A .docx formatted version of the manuscript text (including legends for main figures, EV figures and tables). Please make sure that the changes are highlighted to be clearly visible.
- 2) Individual production quality figure files as .eps, .tif, .jpg (one file per figure).
- 3) A .docx formatted letter INCLUDING the reviewers' reports and your detailed point-by-point responses to their comments. As part of the EMBO Press transparent editorial process, the point-by-point response is part of the Review Process File (RPF), which will be published alongside your paper.
- 4) A complete author checklist, which you can download from our author guidelines (<https://www.embopress.org/page/journal/17574684/authorguide#submissionofrevisions>). Please insert information in the checklist that is also reflected in the manuscript.
- 5) Before submitting your revision, primary datasets produced in this study need to be deposited in

an appropriate public database (see <https://www.embopress.org/page/journal/17574684/authorguide#dataavailability>). Please remember to provide a reviewer password if the datasets are not yet public. The accession numbers and database should be listed in a formal "Data Availability" section (placed after Materials & Method). Please note that the Data Availability Section is restricted to new primary data that are part of this study.

6) We would also encourage you to include the source data for figure panels that show essential data. Numerical data should be provided as individual .xls or .csv files (including a tab describing the data). For blots or microscopy, uncropped images should be submitted (using a zip archive if multiple images need to be supplied for one panel). Additional information on source data and instruction on how to label the files are available at <https://www.embopress.org/page/journal/17574684/authorguide#sourcedata>.

7) Our journal encourages inclusion of *data citations in the reference list* to directly cite datasets that were re-used and obtained from public databases. Data citations in the article text are distinct from normal bibliographical citations and should directly link to the database records from which the data can be accessed. In the main text, data citations are formatted as follows: "Data ref: Smith et al, 2001" or "Data ref: NCBI Sequence Read Archive PRJNA342805, 2017". In the Reference list, data citations must be labeled with "[DATASET]". A data reference must provide the database name, accession number/identifiers and a resolvable link to the landing page from which the data can be accessed at the end of the reference. Further instructions are available at <https://www.embopress.org/page/journal/17574684/authorguide#referencesformat>.

8) We replaced Supplementary Information with Expanded View (EV) Figures and Tables that are collapsible/expandable online. A maximum of 5 EV Figures can be typeset. EV Figures should be cited as 'Figure EV1, Figure EV2' etc... in the text and their respective legends should be included in the main text after the legends of regular figures.

- Additional Tables/Datasets should be labeled and referred to as Table EV1, Dataset EV1, etc. Legends have to be provided in a separate tab in case of .xls files. Alternatively, the legend can be supplied as a separate text file (README) and zipped together with the Table/Dataset file. See detailed instructions here: <https://www.embopress.org/page/journal/17574684/authorguide#expandedview>.

9) The paper explained: EMBO Molecular Medicine articles are accompanied by a summary of the articles to emphasize the major findings in the paper and their medical implications for the non-specialist reader. Please provide a draft summary of your article highlighting

10) For more information: There is space at the end of each article to list relevant web links for further consultation by our readers. Could you identify some relevant ones and provide such information as well? Some examples are patient associations, relevant databases, OMIM/proteins/genes links, author's websites, etc...

11) Every published paper now includes a 'Synopsis' to further enhance discoverability. Synopses are displayed on the journal webpage and are freely accessible to all readers. They include a short stand first (maximum of 300 characters, including space) as well as 2-5 one-sentences bullet points

that summarizes the paper. Please write the bullet points to summarize the key NEW findings. They should be designed to be complementary to the abstract - i.e. not repeat the same text. We encourage inclusion of key acronyms and quantitative information (maximum of 30 words / bullet point). Please use the passive voice. Please attach these in a separate file or send them by email, we will incorporate them accordingly.

Please also suggest a striking image or visual abstract to illustrate your article. If you do please provide a jpeg file 500 px-wide x 400-px high.

12) As part of the EMBO Publications transparent editorial process initiative (see our Editorial at <http://embomolmed.embopress.org/content/2/9/329>), EMBO Molecular Medicine will publish online a Review Process File (RPF) to accompany accepted manuscripts. In the event of acceptance, this file will be published in conjunction with your paper and will include the anonymous referee reports, your point-by-point response and all pertinent correspondence relating to the manuscript. Let us know whether you agree with the publication of the RPF and as here, if you want to remove or not any figures from it prior to publication. Please note that the Authors checklist will be published at the end of the RPF.

I look forward to receiving your revised manuscript.

Yours sincerely,

Lise Roth

Lise Roth, PhD
Editor
EMBO Molecular Medicine

To submit your manuscript, please follow this link:

<https://embomolmed.msubmit.net/cgi-bin/main.plex?el=A2EI6kKK4B5CtbT7I7A9ftdnZyAK7jxZE96RSqNp15rNAY>

***Additional important information regarding Figures**

Photos 400-800 DPI

*Additional important information regarding figures and illustrations can be found at <http://bit.ly/EMBOPressFigurePreparationGuideline>

***** Reviewer's comments *****

Referee #1 (Comments on Novelty/Model System for Author):

Prieto-Garcia and coworkers report USP28 as a deubiquitinase for Δ Np63 which is essential for maintaining lung squamous cancer cell (SCC) identity and its proliferation. Functionally, they show that inhibiting USP28 activity in human cell lines as well as in GEMM mouse models of lung cancer suppresses. The strong point of this work is the use of state of the art GEMM mouse models for lung cancer and molecular characterization of USP28 as a therapeutic vulnerability in SCC. The experiments are in general rigorous and done using state of the art elegant methods. The conclusions are interesting and bring an advance in the field of lung cancer. Here are some aspects of this work that need improvement.

Major comments:

- 1) Authors should attempt to identify the sites of ubiquitination/deubiquitination in DNp63. These information will be useful to further design rescue experiments. In addition, linkage types (K48/K63) of polyubiquitin chains on DNp63 need to be shown by use of proteomics approach.
- 2) The authors should address and/or discuss the issue of inhibition/depletion of USP28 being solely dependent on DNp63 as many other USP28 substrates (Myc, Notch, c-Jun) were known to play a key role in lung cancer progression. They could check for the levels of these oncogenes in their KPL as well as KPLU mouse models and cell lines. In addition, it would be interesting to know what are the effects of USP28 inhibition/depletion in the background of ubiquitination deficient mutants of DNp63?
- 3) The authors should add molecular weight markers for all the blots throughout the manuscript.

Minor comments:

- 1) Immunofluorescence data (Fig.3 A, I) should be moved to supplementary figures. To keep consistency in immunofluorescence data, probe for DNp63 and USP28 as well.
- 2) Please check whether all the figures are cited in the result section. for example. Fig. 3I is not cited.
- 3) Labelling in Fig. 3D is missing. Line no. 478 (figure number missing)
- 4) Why does KPLU panels in Fig. 5C are positive for USP28?

Referee #1 (Remarks for Author):

It is a very dense and elaborate paper with a clear medical importance yet down very carefully at the molecular level and using very innovative mouse models. After revisions this would be appropriate for EMBO Mol Med

Referee #2 (Remarks for Author):

I enjoyed reading Prieto-Garcia and colleagues' work as a pre-print and it has been further improved during the review process to date. It is a significant body of work and a well-written manuscript.

Major comments

- The authors do not discuss prior work showing that USP28 stabilizes TP53 during mitotic quality control which is surprising given the overall structural similarities between TP53 and TP63 and the frequency of TP53 in SCCs. Do they think that USP28i will be affected by TP53 status (most of the

models tested are TP53 deficient)?

- Multiple gene expression analyses (panels 1A, 1B, 4O, 4P, 7B) make comparisons with normal tissue controls but these might be inappropriate in the context of lung SCCs as a result of not being from the matched tissue of origin (i.e. airways instead of lung parenchyma). There is no getting around this but the authors should be more cautious in their interpretation of these data.
- The data concerning tumor initiation are exciting but those on maintenance might fall short of full demonstration that USP28 is required for tumor maintenance. This is not an inducible system, rather cultured cells are instilled into airways from a previous generation of mouse with/without USP28 knockdown. As such, it is possible that the lack of tumors in USP28 knockdown lines reflects impaired engraftment rather than maintenance.

Minor comments

- Line 108 - 'frequently mutated' could be misleading, the authors might specify (i.e. 10% in cSCC, 6% in LUSC etc.)
- Line 259 - the authors could explain why murine USP28 is used here. What is the sequence similarity with the human protein?
- Line 586 - in their discussion the authors state that the USP28 knockdown cells phenocopied deltaNP63 knockdown cells but this is partial (6.8% of upregulated, 21.8% downregulated). More discussion of the possible causes of the discordance would not take away from the value of these findings.
- Figure 4I - what is the statistical test?
- Cut-offs should be defined for all Kaplan-Meier curves - are they split on the median? Upper/lower quartile?
- The numbers of experimental replicates should be added to figure legends throughout; it is often not clear what a blot is representative of or what is being quantified.
- Figure 5D/E - it would be interesting to split tumor size and area by tumor histology here. The possible effects on ADC tumors is underexplored in the data as presented.
- Figure 7G - why do the authors present statistics comparing the shRNAs in different cell lines rather than each to its non-targeting control?
- Figure 8E/H - IC50 values should not be calculated from so few data points. Based on the existing data it is not possible to say where between 10-30 uM the IC50 falls.
- The sources of publicly available datasets should be included in the relevant figure legends.
- Additional information on the cell lines used would be useful (passage numbers/population doublings?), any verification that was performed?
- Methods should be added for the culture of cells from GEMMs as details are currently only provided in a supplementary figure. How long were cells cultured before re-implantation? Did the authors look at the TP63 status of these cells?

Typos: Line 80 (squamous cell carcinomas), 82 (as a consequence), 113 (missing comma), 289 (decreased SCC proliferation), 352, 478 (missing Figure numbers), 493 (delete 'cells').

Referee #3 (Remarks for Author):

The transcription factor Δ Np63 is a master regulator of epithelial cell identity and essential for the survival of squamous cell carcinoma (SCC) of lung, head and neck, oesophagus, cervix and skin. The authors here report that the deubiquitylase USP28 stabilizes Δ Np63 and maintains elevated Δ NP63 levels in SCC by counteracting its proteasome-mediated degradation. Impaired USP28 activity, either genetically or pharmacologically, abrogates the transcriptional identity and suppresses growth and survival of human SCC cells. CRISPR/Cas9-engineered in vivo mouse models establish that endogenous USP28 is strictly required for both induction and maintenance of lung SCC.

These data strongly suggest that targeting Δ Np63 abundance via inhibition of USP28 might constitute a promising strategy for the treatment of SCC tumours.

The paper has clinical relevance for SCC treatment. In general, the experiments are well designed and all controls appropriate. However, the quality of some experiments can be improved. There are some issues/clarifications required to further improve this work.

- 1) The ubiquitination data presented in fig. 2B,C and E (and throughout the paper) are not very easy

to fully understand. Δ Np63 is not clearly marked and it is therefore not easy to appreciate the Δ Np63 ubiquitinated forms. Please add the MW on blot sides and arrows pointing to the correct size for Δ Np63.

- 2) Was Δ Np63 ubiquitination ever tested in reducing condition? This experiment would rule out that changes in ubiquitination are specific for Δ Np63 and not derived from changes in ubiquitination of associated proteins coprecipitating with Δ Np63?
- 3) Is the catalytically inactive USP28C171A working in affecting ubiquitination of endogenous Δ Np63 in SCC cell lines?
- 4) Page 9, line 282-283: "Both SCC cells and tumours depend on Δ Np63 for maintaining proliferation and cell identity" The concept of cell identity is not very clear. Please explain.
- 5) Page 9, line 289: "Depletion of Δ Np63 decreased SCC (Figure S4B) and cell cycle profiling indicated a mild S-phase arrest" Please specify that it is proliferation decreased.
- 6) Page 10, lines 295-305: The rationale of this experiments is not well define as is its conclusion. Do the authors want to prove that the two proteins have partial common function and overlapping control of a defined subset of genes? Please clarify.
- 7) Page 13, Lines 434-435: "Taken together, these data show that USP28 is required for the maintenance of lung SCC" What does it mean maintenance of lung SCC? This concept is not clear. These experiments seem to establish a role in lung metastases.
- 8) Fig. S7D: USP28 is expressed in all SCC cells of different origins. However, Δ Np63 is only expressed in 3 out of 7 cell lines. These data would suggest a role of USP28 which is also Δ Np63-independent. Please discuss this aspect.
- 9) Fig. 8A is partially cut and panel letters not visible. Fig. 8A: adding arrows pointing out active or inactive USP28 would make the panel easier to understand.
- 10) Lines 541-546: "As USP28 has several targets, including the transcription factors cMYC, cJUN and NOTCH (Diefenbacher et al., 2015; Diefenbacher et al., 2014; Popov et al., 2007a; Schulein-Volk et al., 2014), the transcriptional responses in SCC cells are dominated by its effects on Δ Np63, arguing that this axis establishes a unique dependence and hence may open a wide therapeutic window for targeting SCC of various tissue origins via USP28". This section is more appropriate for the Discussion.
- 11) The Discussion is very slim and does not discuss additional targets of USP28 (mentioned above) and their possible contribution to SCC. Please expand considerably.

1st Revision - authors' response

19th Dec 2019

Point-by-point response to reviewers

We would like to thank all three reviewers again for their friendly and thoughtful comments on our manuscript. Please find the detailed replies below.

Referee #1

Prieto-Garcia and coworkers report USP28 as an deubiquitinase for Δ Np63 which is essential for maintaining lung squamous cancer cell (SCC) identity and its proliferation. Functionally, they show that inhibiting USP28 activity in human cell lines as well as in GEMM mouse models of lung cancer suppresses. The strong point of this work is the use of state of the art GEMM mouse models for lung cancer and molecular characterization of USP28 as a therapeutic vulnerability in SCC. The experiments are in general rigorous and done using state of the art elegant methods. The conclusions are interesting and bring an advance in the field of lung cancer. Here are some aspects of this work that need improvement.

Major comments:

1) Authors should attempt to identify the sites of ubiquitination/deubiquitination in DNp63. These information will be useful to further design rescue experiments. In addition, linkage types (K48/K63) of polyubiquitin chains on DNp63 need to be shown by use of proteomics approach.

We appreciate the reviewers' comment and suggestion regarding the ubiquitylation status of DNp63, and which sites are used. To address which lysine residue and what kind of chain type is present on DNp63 under various conditions, we collaborated with experts in the field of proteomics.

In order to analyse the ubiquitylation pattern found on DNp63, a total of 12 conditions were analysed by mass spectrometry:

- 1- Non targeting control IP DNp63
- 2- shRNA-USP28 IP DNp63

- 3- Non targeting control TUBE IP DNp63
- 4- shRNA-USP28 TUBE-IP DNp63

- 5- Non targeting control MG132 IP DNp63
- 6- shRNA-USP28 MG132 IP DNp63

- 7- DMSO TUBE IP DNp63
- 8- AZ1 IP DNp63
- 9- AZ1 MG132 IP DNp63
- 10- AZ1 TUBE IP DNp63

- 11- DMSO IP DNp63
- 12- MG132 IP DNp63

All samples were measured on a QExactive HF with a 60min gradient. For data analysis all RAW-files were combined and searched against human proteome database (SwissProt: 2018-11-21) with diGly as dynamic modification at lysines. As expected, in all samples we were able to detect DNp63 with an overall sequence coverage of ~60%.

Protein Sequence Coverage

```

M L Y L E N N A Q T Q F S E P Q Y T N L G L L N S M D Q Q I Q N G S S S T S P Y N T D H A Q N S V T A P S P Y A Q P S S T F D A L S P S P A I
P S N T D Y P G P H S F D V S F Q S S T A K S A T W T Y S T E L K K L Y C Q I A K T C P I Q I K V M T P P P
Q G A V I R A M P V Y K K A E H V T E V V K R C P N H E L S R E F N E G Q I A P P S H L I R V E G N S H A Q Y V E D P I T G R Q S V L V P Y
E P P Q V G T E F T T V L Y N F M C N S S C V G G M N R R P I L I I V T L E T R D G Q V L G R R C F E A R I C A C P G R D R K A E D S I R
K Q Q V S D S T K N G D G T K R P F R Q N T H G I Q M T S I K K R R S P D D E L L Y L P V R G R E T Y E M L L K I K E S L E L M Q Y L P Q H
T I E T Y R Q Q Q Q
Q Q H Q H L L Q K Q T S I Q S P S S Y G N S S P P L N K M N S M N K L P S V S Q L I N P Q Q R N A L T P T T I P D G M G A N I P M M G T H M
P M A G D M N G L S P T Q A L P P P L S M P S T S H C T P P P P Y P T D C S I V S F L A R L G C S S C L D Y F T T Q G L T T I Y Q I E H Y S
M D D L A S L K I P E Q F R H A I W K G I L D H R Q L H E F S S P S H L L R T P S S A S T V S V G S S E T R G E R V I D A V R F T L R Q T I
S F P P R D E W N D
F N F D M D A R R N K Q Q R I K E E G E

```

(Detected peptides are highlighted in green)

We could not detect any GG remnants on p63, which might be the result of low stoichiometry of the ubiquitylated residues compared to the unmodified. Additionally, we cannot exclude unfavourable physiochemical properties of the modified peptide for mass spectrometry.

In parallel, by performing SPOT interaction studies, we were able to detect two modes of interaction for USP28 on DNp63; interaction via the phospho-degron motive of Fbxw7, or via lysine containing domains (Figure S2, Figure for reviewers 1 and lines 865 to 897). Therefore, we decided to generate DNp63^{KtoR} and DNp63^{Kless} (deletion of lysine encoding codons) mutant constructs (Figure S7A and B) as tools to rescue the USP28 dependency and were able to observe that treatment with the USP28 inhibitor did not reduce mutant protein abundance (Figure S7B). This is in line with the previous observations that the mammalian transcription factor MYC, which harbours several lysine residues, is still degraded in a proteasomal fashion when lysines are available (Jaenicke et al., 2016). Hence, all lysine residues were required to be mutated to prevent degradation.

To address linkage types on DNp63 in cells, we performed Ni-NTA pulldown experiments under denaturing conditions in HEK293 cells transiently transfected with either His-K48 or His-K63 only ubiquitin and FLAG-tagged DNp63. In this experimental set up both chain types could be identified on DNp63 (please see Figure 2C), but only His-K48 ubiquitin was removed when USP28 was overexpressed, showing that USP28 removes degradative ubiquitin chains on DNp63, resulting in the stabilisation of this oncoprotein.

We also attempted to identify ubiquitin chains on endogenous DNp63, but, unfortunately, we were not able to detect the ubiquitin chain types found on endogenous DNp63 by proteomic approaches.

2) The authors should address and/or discuss the issue of inhibition/depletion of USP28 being solely dependent on DNp63 as many other USP28 substrates (Myc, Notch, c-Jun) were known to play a key role in lung cancer progression. They could check for the levels of these oncogenes in their KPL as well as KPLU mouse models and cell lines. In addition, it would be interesting to know what are the effects of USP28 inhibition/depletion in the background of ubiquitination deficient mutants of DNp63?

In other tissue models such as small intestine and colon, cJUN, cMYC and Notch1 are required for tumour growth (Diefenbacher et al., 2015; Diefenbacher et al., 2014; Jaenicke et al., 2016; Schulein-Volk et al., 2014). We decided to address this point by performing immunohistochemistry on lung tumour samples of our CRISPR/Cas9 mouse cohort, KPL and KPLU, and complemented the analysis by assessing the protein abundance of the USP28 substrates in tumour lysates from KPL and KPLU tumour explants (please see Figure 5F-G and S5G). In line with our observation in USP28^{shRNA} and DNp63^{shRNA} cell lines (Figure

S4J), tumours depleted of USP28 showed reduced protein levels of DNp63, but also of active Notch1 by immunoblotting and IHC, while protein levels of cMYC and cJun were not affected (Figure 5G and S5G). We added a subsection within the discussion part focusing on a potential involvement of NOTCH signaling on the observed phenotypes.

New Figure 5F and G, and S5G (please see manuscript and figures for more information).

To address the effects of inhibition of USP28 in ubiquitin deficient DNp63, we generated two mutant constructs either exchanging lysine to arginine (DNp63^{KtoR}) or depleting all lysine residues (DNp63^{Kless}). We used the Fbxw7 and USP28 wild type cell line HEK293T to transiently overexpress the two mutant constructs. As a control we overexpressed the wild type form of DNp63. We were able to observe an increase in protein expression of the two mutant variants of DNp63, when compared to the wild type variant, however, these constructs resulted in a induced expression of cytokeratin 14, an SCC identity marker directly regulated by DNp63. Furthermore, addition of the inhibitor AZ1 resulted in the reduction of wild type DNp63 and cytokeratin 14, while the mutant variants DNp63^{KtoR} and DNp63^{Kless} were not affected (Figure S7). This demonstrates that USP28 regulates DNp63 protein stability and this process requires the presence of lysine residues on the substrate. Additionally, the physiological role of DNp63 depends on lysine residues as well.

New Figure S7A and B (please see manuscript and figures for more information).

3) *The authors should add molecular weight markers for all the blots throughout the manuscript.*

We are sorry for this mistake and have added molecular markers where suitable.

Minor comments:

1) *Immunofluorescence data (Fig. 3 A, I) should be moved to supplementary figures. To keep consistency in immunofluorescence data, probe for DNp63 and USP28 as well.*

Following the reviewers' suggestion, we have moved the immunofluorescence staining to the corresponding supplementary figure (Figure S3).

2) *Please check whether all the figures are cited in the result section. for example, Fig. 3I is not cited.*

We are sorry for this mistake and have addressed figure citation errors throughout the manuscript.

3) *Labelling in Fig. 3D is missing. Line no. 478 (figure number missing)*

We thank the reviewer for noticing this error, which has been amended, and labeling was added to the figure.

4) *Why does KPLU panels in Fig. 5C are positive for USP28?*

As USP28 protein abundance is significantly reduced in tumour lysates from KPLU mice, when compared to KPL tumour samples, we suggest that tumours arising in the KPLU lung cancer model system are only partially depleted for USP28. To address this possibility *in vitro*, we attempted to CRISPR/Cas9 deplete USP28, using the same sgRNA cassettes from the KPLU vector, in established primary tumour cell lines from the KPL mouse model, but failed to do so. Targeting Usp28 had detrimental effects on cell survival in our model systems and primary cell lines. Therefore, we speculate that the remaining staining is due to incomplete targeting of USP28, and this incomplete event enables tumour growth.

Referee #1 (Remarks for Author):

It is a very dense and elaborate paper with a clear medical importance yet down very carefully at the molecular level and using very innovative mouse models. After revisions this would be appropriate for EMBO Mol Med

Referee #2 (Remarks for Author):

I enjoyed reading Prieto-Garcia and colleagues' work as a pre-print and it has been further improved during the review process to date. It is a significant body of work and a well-written manuscript.

Major comments

- *The authors do not discuss prior work showing that USP28 stabilizes TP53 during mitotic quality control which is surprising given the overall structural similarities between TP53 and TP63 and the frequency of TP53 in SCCs. Do they think that USP28i will be affected by TP53 status (most of the models tested are TP53 deficient)?*

We thank the reviewer for the positive evaluation of our study, and raising this point. Our findings regarding the role of Tp53 are presented in the figure below. We have also added a subsection in the Discussion paragraph regarding our findings in context to TP53. Please see lines 670 to 682.

It was reported that TP53 stability was affected in a USP28-TP53BP1-cell cycle dependent fashion (Fong et al., 2016; Lambrus et al., 2016; Meitinger et al., 2016). *In vivo*, however, using a whole body acute depletion model ($APC^{min\Delta/+}; Usp28^{fl/fl}; Rosa26^{Sor-CreERT2} +/-$ Tamoxifen) did not result in changes of endogenous TP53, or in its activation, but affected tumour growth (Diefenbacher et al., 2014). Also, *in vivo* germ line depletion of *Usp28* was well tolerated and did not induce classic *Tp53*^{-/-} phenotypes, like lymphoma or thymoma formation, in these mouse cohorts (Schulein-Volk et al., 2014).

To assess if blocking USP28 via AZ1 alters TP53 protein abundance, we treated HEK293 cells for 48 hours with the inhibitor and analysed Tp53 protein abundance by immunoblotting. As a control for the treatment, cells

were transiently transfected with FLAG-tagged DNp63. While DNp63 protein abundance was decreased in the presence of AZ1, endogenous TP53 showed no alteration, hence AZ1 was able to affect DNp63 protein stability irrespective of TP53.

Next, we assessed the TP53 status in the cell lines used in this study, as well analysed the mutation frequency in NSCLC ADC and SCC (Figure for reviewers 2). The human cancer cell lines were either TP53 mutant (classic hot spot areas) or were transformed by HPV, hence, TP53 inactivated. In lung cancer, in particular SCC, but also in the other tumour entities analysed, the majority of patients' harbor mutations in TP53. This led us to the decision to include the sgRNA targeting Tp53 also in our mouse model.

To answer the question if Usp28 inhibition will be efficient in Tp53 wild type tumour cells, at this point one can only speculate. Previous mouse models analyzing and targeting Usp28 genetically were performed in a Tp53 wild type background. Loss of Usp28 affected tumour growth and extended life expectancy, at least *in vivo* in colon cancer models (Diefenbacher et al., 2014). In the future, addressing the ability of USP28 inhibition in Tp53 wild type tumours is interesting and will be considered by our laboratory in upcoming *in vivo* studies.

We have added a section regarding TP53 and USP28 in the discussion. Please see lines 670 to 682.

• Multiple gene expression analyses (panels 1A, 1B, 4O, 4P, 7B) make comparisons with normal tissue controls but these might be inappropriate in the context of lung SCCs as a result of not being from the matched tissue of origin (i.e. airways instead of lung parenchyma). There is no getting around this but the authors should be more cautious in their interpretation of these data.

We share the reviewers' opinion that It is a common issue with public available datasets. We also agree with the reviewer that it is unclear from which tissue subtype 'Normal' and 'Tumor' arise, e.g. non-transformed alveolar tissue would be the suitable 'Normal' tissue control for adenocarcinoma as 'Tumor' sample.

Therefore, we added the information what groups were compared within the figure legends.

• The data concerning tumor initiation are exciting but those on maintenance might fall short of full demonstration that USP28 is required for tumor maintenance. This is not an inducible system, rather cultured cells are instilled into airways from a previous generation of mouse with/without USP28 knockdown. As such, it is possible that the lack of tumors in USP28 knockdown lines reflects impaired engraftment rather than maintenance.

To address the reviewers comment, we decided to merge the tumour initiation data with the tumour transplant dataset and, as a consequence, renamed this paragraph to tumour initiation and engraftment to reflect this change. Since the first in class pharmacologic inhibitor AZ1 is available, we decided to tackle the interesting topic of tumour maintenance by combining

the isogenic transplant model with AZ1 treatment in vivo. Thereby, we were able to test if established tumours show any signs of response upon treatment with the inhibitor. Please see the new Figure 8 and Figure S8. In brief, DNp63 expressing KPL tumour cells readily engrafted in wild type C57BL/6J mice and formed primary tumours. 28 days post intratracheal transplant, mice were treated with either vehicle or two concentrations of AZ1, and tumour burden was analysed at end point. AZ1 treated mice showed smaller lesions and reduced DNp63 protein abundance, as well inactivated USP28, as assessed by ubiquitin suicide probes. In vivo application of AZ1 resulted in an overall reduction of tumour load and affected tumour growth/maintenance.

New Figure 8 (excluding the model and supplementary data. Please see manuscript and figures for more information).

Minor comments

- Line 108 - 'frequently mutated' could be misleading, the authors might specify (i.e. 10% in cSCC, 6% in LUSC etc.)

To avoid confusion regarding the mutational frequency of Fbxw7 in human tumours, we have specified the occurring frequency in the text

based on data available via *cbioportal.org* (Figure for reviewers 3 and line 108-110).

- *Line 259 - the authors could explain why murine USP28 is used here. What is the sequence similarity with the human protein?*

Human and murine USP28 share a high degree of structural similarity. We have included a chart highlighting the degree of conservation in structure among different species (Figure S3F and Figure for reviewers 4). These constructs were readily available at our department and therefore used in the study.

- *Line 586 - in their discussion the authors state that the USP28 knockdown cells phenocopied deltaNP63 knockdown cells but this is partial (6.8% of upregulated, 21.8% downregulated). More discussion of the possible causes of the discordance would not take away from the value of these findings.*

As shown in our study (Figure 4), USP28 and DNp63 share a commonly regulated gene signature specific for SCC. However, USP28 regulates additional proto-oncogenes and we also detected changes in NOTCH signaling in USP28 knock down cells, while JUN/AP-1 and MYC were not affected. Furthermore, genetic or pharmacologic targeting of USP28 does not fully recapitulate the genetic depletion of the essential SCC transcription factor DNp63, but showed a high level of overlap with regard to basic genetic processes being affected. Therefore, targeting USP28 is a good surrogate target for affecting DNp63 function in SCC. Additionally, we included a section in the discussion mentioning transcription factors regulated by USP28, which are also essential for tumour formation and maintenance, but have a lesser importance in SCC. Please see lines lines 684 to 691 and 726 to 737.

- *Figure 4I - what is the statistical test?*

We have amended this for all figures presented and included it in the figure legends. In Figure 4I we used a using two-tailed t-test.

- *Cut-offs should be defined for all Kaplan-Meier curves - are they split on the median? Upper/lower quartile?*

We assessed the cut offs of the presented Kaplan-Meier curves following the pre-sets of the used online available tools, hence, *>all possible cutoff values between the lower and upper quartiles are computed, and the best performing threshold is used as a cutoff. The results page will display the False Discovery Rate in addition to the p-value< (quote kmplot.com)*. To highlight how the expression and survival data presented in our study was generated, we have generated the Figure for reviewers 5. Here, we show patient survival and thresholding for each dataset side-by-side. When this checkbox is selected, all possible cutoff values between the lower and

upper quartiles are computed, and the best performing threshold is used as a cutoff. The results page will display the False Discovery Rate in addition to the p-value.

- *The numbers of experimental replicates should be added to figure legends throughout; it is often not clear what a blot is representative of or what is being quantified.*

We have amended this for all figures presented and included it in the figure legends.

- *Figure 5D/E - it would be interesting to split tumor size and area by tumor histology here. The possible effects on ADC tumors is underexplored in the data as presented.*

The requested data is now presented in Figure S5F. As previously reported, the combinatorial loss of Lkb1/STK11 with Tp53 and mutation of Kras(to KrasG12D) results in the development of ADC and SCC, respective to marker expression (TTF1+/DNp63-/CK5- vs TTF1-/DNp63+/CK5+)(Han et al., 2014). Loss of USP28 by CRISPR/Cas9 during tumour initiation resulted in a significant reduction of overall tumour burden, and only ADC marker expressing tumours were detected.

New Figure S5F (Please see manuscript and figures for more information).

- *Figure 7G - why do the authors present statistics comparing the shRNAs in different cell lines rather than each to it's non-targeting control?*

We thank the reviewer for addressing the statistic testing and presentation of this comparative experiment. The reason for comparing the respective shRNA of different cell lines was to highlight the observed dependency for maintaining high levels of USP28, in particular in the SCC lines. ADC tumour cells showed a weak anti-proliferative effect, most likely due to reduced levels of NOTCH signalling, but SCC lines exhibited a strong anti-proliferative effect.

- *Figure 8E/H - IC50 values should not be calculated from so few data points. Based on the existing data it is not possible to say where between 10-30 μ M the IC50 falls.*

To address this point raised by the reviewer, we included additional concentrations and reassessed cell numbers. This led to a change in the estimated IC₅₀ values and is now documented in the manuscript (Figure 7).

- *The sources of publicly available datasets should be included in the relevant figure legends.*

We thank the reviewer to highlight the missing information. We have amended this for all figures presented and included it in the figure legends as well as Material and Methods.

- *Additional information on the cell lines used would be useful (passage numbers/population doublings?), any verification that was performed?*

We thank the reviewer to highlight the missing information. We have performed SNP profiling of the parental cell lines used in our study. The results are attached as supplementary information for reviewers. The human lung cancer cell line LUDLU-1 grows as a semi-attached cell line. To ease biochemical studies, we decided to select for an adherent clone. This clone presented an altered STR profile. Therefore, we decided to rename the cell line used in our study to avoid confusion with the parental cell line to LUDLU-1^{adh} and added a section in the Material and Methods part of the manuscript, highlighting these differences.

> The human lung cancer cell line LUDLU-1 is maintained as a semi-attached/floating culture. We subjected this cell line to a selection process enriching for an adherent clone, which was further propagated and used for all experiments within this study. Therefore, to highlight the difference towards the parental cell line, we decided to mark this by adding the suffix adh., hence naming the cell line LUDLU-1^{adh}. The selection process did not alter the expression of endogenous Δ Np63 and SCC markers. It is noteworthy that the overall STR profile of the created subclone is similar, but not identical, to the parental cell line. (Please see STR profiles in supplementary data). < (Please see lines 778 to 785 and STR profiles for reviewers).

- *Methods should be added for the culture of cells from GEMMs as details are currently only provided in a supplementary figure. How long were cells cultured before re-implantation? Did the authors look at the TP63 status of these cells?*

We thank the reviewer to draw our attention to this missing section in the Material and Methods part of the manuscript. A detailed protocol was added describing the primary culture and establishing of murine lung tumour cell lines. Please see lines 800 to 817. The DNp63 status of the re-transplanted cell lines was assessed using immunofluorescence staining. In

Figure 8 we compare the DNp63 expression of a KP and a KPL cell clone by using the clinically used anti-p63 clone 4A4 from Ventana. This KPL clone was also used for the transplant experiment shown in Figure 8.

Typos: Line 80 (squamous cell carcinomas), 82 (as a consequence), 113 (missing comma), 289 (decreased SCC proliferation), 352, 478 (missing Figure numbers), 493 (delete 'cells').

We thank the reviewer for highlighting these mistakes. We corrected the passages mentioned in the manuscript text.

Referee #3 (Remarks for Author):

The transcription factor Δ Np63 is a master regulator of epithelial cell identity and essential for the survival of squamous cell carcinoma (SCC) of lung, head and neck, oesophagus, cervix and skin. The authors here report that the deubiquitylase USP28 stabilizes Δ Np63 and maintains elevated Δ NP63 levels in SCC by counteracting its proteasome-mediated degradation. Impaired USP28 activity, either genetically or pharmacologically, abrogates the transcriptional identity and suppresses growth and survival of human SCC cells. CRISPR/Cas9-engineered in vivo mouse models establish that endogenous USP28 is strictly required for both induction and maintenance of lung SCC.

These data strongly suggest that targeting Δ Np63 abundance via inhibition of USP28 might constitute a promising strategy for the treatment of SCC tumours.

The paper has clinical relevance for SCC treatment. In general, the experiments are well designed and all controls appropriate. However, the quality of some experiments can be improved.

There are some issues/clarifications required to further improve this work.

1) The ubiquitination data presented in fig. 2B,C and E (and throughout the paper) are not very easy to fully understand. Δ Np63 is not clearly marked and it is therefore not easy to appreciate the Δ Np63 ubiquitinated forms. Please add the MW on blot sides and arrows pointing to the correct size for Δ Np63.

We thank the reviewer for his positive evaluation of our study. We addressed the difficulty to see changes in the ubiquitin pattern by adding molecular weight markers throughout the study and marking the mono-ubiquitinated forms of DNp63.

2) Was Δ Np63 ubiquitination ever tested in reducing condition? This experiment would rule out that changes in ubiquitination are specific for Δ Np63 and not derived from changes in ubiquitination of associated proteins coprecipitating with Δ Np63?

In our study, ubiquitylation-experiments carried out in HEK293 cells (Figure 2 and S2) were performed by overexpression of His-tagged

ubiquitin, followed by Ni-NTA pulldown in the presence of 6M Guanidinium and β -Mercaptoethanol, followed by immunoblotting against DNP63. Thereby, analyzed ubiquitin modifications are specific to DNP63. Subsequent ubiquitylation experiments were performed in non-denaturing and conditions using TUBE pulldown assays. We recognized that a dedicated section for ubiquitin pulldown experiments is missing from the Material and Methods section. This has been amended. Please see lines 899 to 925.

3) Is the catalytically inactive USP28^{C171A} working in affecting ubiquitination of endogenous Δ Np63 in SCC cell lines?

To address this question, we transiently transfected A431 cells with USP28^{C171A} and investigated total DNP63 protein abundance by western blot, as well its ubiquitylation by TUBE (Figure S3J and K). In transiently transfected cells, overexpression of the catalytic inactive form of USP28 led to a minor reduction of endogenous DNP63, as well facilitated an increase in its ubiquitylation. Since USP28 is known to homo-dimerize (Gersch et al., 2019; Sauer et al., 2019), the catalytic dead isoform may dimerise with endogenous USP28, thereby functioning as a dominant negative mutant. This speculation has also been added to the manuscript. Please see lines 706 to 709.

New Figure S3J and K (Please see manuscript and figures for more information).

4) Page 9, line 282-283: "Both SCC cells and tumours depend on Δ Np63 for maintaining proliferation and cell identity" The concept of cell identity is not very clear. Please explain.

We appreciate the reviewer for addressing the issue of defining cell identity. In SCC it refers to the maintained expression of lineage markers of keratinization, such as CK5 and 14, which are otherwise absent from tumours, in particular ADC. This section has been added to the manuscript. Please see lines 301 to 304.

5) Page 9, line 289: "Depletion of Δ Np63 decreased SCC (Figure S4B) and cell cycle profiling indicated a mild S-phase arrest" Please specify that it is proliferation decreased.

We changed the text to highlight that loss of DNp63 or USP28 affected cell proliferation and resulted in an accumulation of cells in S-phase. Please see lines 309 to 310.

6) Page 10, lines 295-305: The rationale of this experiments is not well define as is its conclusion.

Do the authors want to prove that the two proteins have partial common function and overlapping control of a defined subset of genes? Please clarify.

We understand the confusion caused by the choice of wording. To address this point we rewrote the conclusion of this paragraph to clarify our observation and interpretation of the data. In brief, by using genetic depletion experiments, we demonstrated that USP28 regulates classic SCC gene expression profiles via its ability to control DNp63 protein abundance. Hence, cells depleted for USP28 or DNp63 show a partial, but essential overlap in SCC specific genes being downregulated. Please see lines 383 to 388.

7) Page 13, Lines 434-435: "Taken together, these data show that USP28 is required for the maintenance of lung SCC" What does it mean maintenance of lung SCC? This concept is not clear. These experiments seem to establish a role in lung metastases.

We kindly would like to refer to point 3 of reviewer 2.

> We acknowledge the concerns raised by the reviewer and decided to merge the tumour initiation data with the tumour transplant dataset and, as a consequence, renamed this paragraph to tumour initiation and engraftment to reflect this change. Since the first in class pharmacologic inhibitor AZ1 is available, we decided to tackle the interesting topic of tumour maintenance by combining the isogenic transplant model with AZ1 treatment in vivo. Thereby, we were interested if established tumours show any signs of response upon treatment with the inhibitor. Please see the new Figure 8 and Figure S8. In brief, DNp63 expressing KPL tumour cells readily engrafted in wild type C57BL/6J mice and formed primary tumours. 28 days post intratracheal transplant, mice were treated with either vehicle or two concentrations of AZ1, and tumour burden was analysed at end point. AZ1 treated mice showed smaller lesions and reduced DNp63 protein abundance, as well inactivated USP28, as assessed by warhead/ubiquitin suicide probes. In vivo application of AZ1 resulted in an overall reduction of tumour load and affected tumour growth/maintenance<

8) Fig. S7D: USP28 is expressed in all SCC cells of different origins. However, Δ Np63 is only expressed in 3 out of 7 cell lines. These data would suggest a role of USP28 which is also Δ Np63-independent. Please discuss this aspect.

We thank the reviewer for drawing our attention to this issue. The labeling used in the current draft led to the impression that all shown cell lines are of SCC origin. In fact, the cell lines shown in Figure S7D (see

below) represent western blots of ADC and SCC cell lines of various origins. SCC lines are marked red, the corresponding ADC in blue. It is note mentioning that the SCC line SiHa does not express detectable levels of DNp63. Therefore, the major effect of USP28 inhibition is observed in cell lines expressing DNp63, however, interfering with USP28 activity also affects the proliferation of ADC, but to a lesser extent. Therefore, one cannot exclude an DNp63 independent effect in ADC. Based on our findings, we have included a discussion on a possible role of USP28 in the regulation of NOTCH signaling in KRas driven ADC tumours. This has been added to the discussion. Please see lines 726 to 737.

9) Fig. 8A is partially cut and panel letters not visible. Fig. 8A: adding arrows pointing out active or inactive USP28 would make the panel easier to understand.

New Figure 7A (Please see manuscript and figures for more information).

10) Lines 541-546: "As USP28 has several targets, including the transcription factors cMYC, cJUN and NOTCH (Diefenbacher et al., 2015; Diefenbacher et al., 2014; Popov et al., 2007a; Schulein-Volk et al., 2014), the transcriptional responses in SCC cells are dominated by its effects on ΔNp63, arguing that this axis establishes a unique dependence and hence may open a wide therapeutic window for targeting SCC of various tissue origins via USP28". This section is more appropriate for the Discussion.

The section has been rewritten and subsequently moved to the discussion part of the manuscript. Please see lines 726 to 737.

11) The Discussion is very slim and does not discuss additional targets of

USP28 (mentioned above) and their possible contribution to SCC. Please expand considerably.

We thank the reviewer for this suggestion and we have expanded the discussion section by including points raised by the reviewers.

Literature index

- Diefenbacher, M.E., Chakraborty, A., Blake, S.M., Mitter, R., Popov, N., Eilers, M., and Behrens, A. (2015). Usp28 counteracts Fbw7 in intestinal homeostasis and cancer. *Cancer Res* 75, 1181-1186.
- Diefenbacher, M.E., Popov, N., Blake, S.M., Schulein-Volk, C., Nye, E., Spencer-Dene, B., Jaenicke, L.A., Eilers, M., and Behrens, A. (2014). The deubiquitinase USP28 controls intestinal homeostasis and promotes colorectal cancer. *J Clin Invest* 124, 3407-3418.
- Fong, C.S., Mazo, G., Das, T., Goodman, J., Kim, M., O'Rourke, B.P., Izquierdo, D., and Tsou, M.F. (2016). 53BP1 and USP28 mediate p53-dependent cell cycle arrest in response to centrosome loss and prolonged mitosis. *Elife* 5.
- Gersch, M., Wagstaff, J.L., Toms, A.V., Graves, B., Freund, S.M.V., and Komander, D. (2019). Distinct USP25 and USP28 Oligomerization States Regulate Deubiquitinating Activity. *Mol Cell* 74, 436-451 e437.
- Han, X., Li, F., Fang, Z., Gao, Y., Li, F., Fang, R., Yao, S., Sun, Y., Li, L., Zhang, W., *et al.* (2014). Transdifferentiation of lung adenocarcinoma in mice with Lkb1 deficiency to squamous cell carcinoma. *Nat Commun* 5, 3261.
- Jaenicke, L.A., von Eyss, B., Carstensen, A., Wolf, E., Xu, W., Greifenberg, A.K., Geyer, M., Eilers, M., and Popov, N. (2016). Ubiquitin-Dependent Turnover of MYC Antagonizes MYC/PAF1C Complex Accumulation to Drive Transcriptional Elongation. *Mol Cell* 61, 54-67.
- Lambrus, B.G., Daggubati, V., Uetake, Y., Scott, P.M., Clutario, K.M., Sluder, G., and Holland, A.J. (2016). A USP28-53BP1-p53-p21 signaling axis arrests growth after centrosome loss or prolonged mitosis. *J Cell Biol* 214, 143-153.
- Meitinger, F., Anzola, J.V., Kaulich, M., Richardson, A., Stender, J.D., Benner, C., Glass, C.K., Dowdy, S.F., Desai, A., Shiau, A.K., *et al.* (2016). 53BP1 and USP28 mediate p53 activation and G1 arrest after centrosome loss or extended mitotic duration. *J Cell Biol* 214, 155-166.
- Sauer, F., Klemm, T., Kollampally, R.B., Tessmer, I., Nair, R.K., Popov, N., and Kisker, C. (2019). Differential Oligomerization of the Deubiquitinases USP25 and USP28 Regulates Their Activities. *Mol Cell* 74, 421-435 e410.
- Schulein-Volk, C., Wolf, E., Zhu, J., Xu, W., Taranets, L., Hellmann, A., Jaenicke, L.A., Diefenbacher, M.E., Behrens, A., Eilers, M., *et al.* (2014). Dual regulation of Fbw7 function and oncogenic transformation by Usp28. *Cell Rep* 9, 1099-1109.

Figures for reviewers

Figure 1
A

B

Output: www.ubpred.org

Residue	Score	Ubiquitinated
94	0.78	Yes Medium confidence
105	0.31	No
106	0.26	No
113	0.27	No
120	0.41	No
138	0.23	No
139	0.18	No
148	0.56	No
259	0.74	Yes Medium confidence
267	0.93	Yes High confidence
275	0.49	No
281	0.65	Yes Low confidence
297	0.68	Yes Low confidence
298	0.68	Yes Low confidence
322	0.47	No
324	0.52	No
355	0.85	Yes High confidence
374	0.77	Yes Medium confidence
380	0.73	Yes Medium confidence
494	0.43	No
505	0.48	No
577	0.80	Yes Medium confidence
582	0.84	Yes High confidence

LOW, MEDIUM and HIGH confidence

C

csb.cse.yzu.edu.tw/UbiSite/index.php

Order	Protein Name	Locations	Score (Confidence)	Ubiquitination Sites	Substrate Motifs
1	ΔNP63	94	0.7778 (Low)	LDQETA K KAKWY	K...E...
2	ΔNP63	105	0.26475 (Low)	VTEEL K KLTQI	K...E...
3	ΔNP63	106	0.26011 (Low)	VTEEL K LKQTA	K...E...
4	ΔNP63	138	0.27761 (Low)	ADQYK K KAKWY	K...E...
5	ΔNP63	139	0.43315 (Low)	ADQYK K ADQYK	K...E...
6	ΔNP63	259	0.24658 (Low)	YDQYK K ADQYK	K...E...
7	ΔNP63	267	0.43274 (Low)	YDQYK K QDQYK	K...E...
8	ΔNP63	275	0.39943 (Low)	YDQYK K NQDQYK	K...E...
9	ΔNP63	281	0.28718 (Low)	YDQYK K KQDQYK	K...E...
10	ΔNP63	297	0.33358 (Low)	YDQYK K NQDQYK	K...E...
11	ΔNP63	298	0.26509 (Low)	YDQYK K NQDQYK	K...E...
12	ΔNP63	322	0.22494 (Low)	VTEEL K KAKWY	K...E...
13	ΔNP63	324	0.43274 (Low)	VTEEL K KAKWY	K...E...
14	ΔNP63	355	0.43315 (Low)	YDQYK K QDQYK	K...E...
15	ΔNP63	380	0.28643 (Low)	YDQYK K LKQYK	K...E...
16	ΔNP63	494	0.39378 (Low)	YDQYK K LKQYK	K...E...
17	ΔNP63	505	0.264819 (Low)	YDQYK K QDQYK	K...E...

HIGH confidence or SCORE > 0,5

D

Supplementary Figure for reviewers: mapping USP28 binding on ΔNP63 via SPOT array
 A) Immunoblot of endogenous USP28 (derived from A-431 cells) binding to various domains of ΔNP63, and detected by immunofluorescence staining by using an anti-USP28 antibody derived in rabbit, followed by incubation with an anti-rabbit antibody coupled to ALEXA-800, derived in donkey. Protein binding was imaged using the LiCOR Clx system.
 B) computational prediction of propable ubiquitylation of various lysine residues of ΔNP63 via the free online tool www.ubiprep.org
 C) computational prediction of propable ubiquitylation of various lysine residues of ΔNP63 via the free online tool csb.cse.yzu.edu.tw/UbiSite/index.php
 D) highlighted USP28 binding areas identified via the SPOT array within ΔNP63. Several lysine containing domains and the Fbxw7 phospho-degion motif are bound by USP28

Figure 2

Supplementary Figure for reviewers: TP53 sensitivity to AZ1 treatment and status in human tumours

A) immunoblot against Δ Np63 and TP53 of HEK293 cells transiently transfected with FLAG-tagged Δ Np63 and either control or AZ1 treated for 48 hours. Actin served as loading control.

B) TP53 mutational status in cell lines used in this study. Data generated with p53.iarc.fr/celllines.aspx

C) TP53 mutational status in human NSCLC, ADC and SCC. Data generated with cbioportal.org

D) TP53 mutational status in human non-small cell lung cancer, head-and-neck, oesophagus and cervix. Data generated with cbioportal.org

Figure 3

Supplementary Figure for reviewers: Fbxw7 is frequently mutated in human SCC

Analysis of occurring Fbxw7 genetic alterations in lung squamous (LUSC), cervical (CESC), oesophagus (ESCA), head and neck (HNSC) tumours. <https://www.cbioportal.org>

Figure 4

Supplementary Figure for reviewers: USP28 homology

Analysis of sequence homology between USP28 and additional DUBs of various animals using the online tool www.ebi.ac.uk

- A) Protein sequence similarities of USP28 and USP25 between human, mouse, rat and chicken, relative to human USP28
- B) phylogenetic tree of DUBs showing clustering of USP28/USP25 of various organisms
- C) FASTA alignment of the protein sequence of deubiquitylases of various organisms. Colour indicates significance of similarity.

Figure 5

KM PLOT
FIG.1

Cutoff value used
in analysis: 254

Cutoff value used
in analysis: 1402

Cutoff value used
in analysis: 373

R2
FIG.S7

STR profiles of cell lines used in this study:

Zelllinie H1299		
Marker	Vergleichsprofil	H1299
Amelogenin	X	X
D3S1358	n.a.	17
D1S1656	n.a.	12
D6S1043	n.a.	11
D13S317	12	12
Penta E	n.a.	11
D16S539	12/13	12/13
D18S51	n.a.	16
D2S1338	n.a.	23/24
CSF1PO	12	12
Penta D	n.a.	13
THO1	6/9.3	6/9.3
VWA	16/17/18	16/17/18
D21S11	n.a.	32.2
D7S820	10	10
D5S818	11	11
TPOX	8	8
D8S1179	n.a.	10/(12)/13
D12S391	n.a.	21/22
D19S433	n.a.	14
FGA	n.a.	20

Zelllinie Detroit		
Marker	Vergleichsprofil	Detroit
Amelogenin	X	X
D3S1358	n.a.	15/16
D1S1656	n.a.	15.3/17.3
D6S1043	n.a.	12/18
D13S317	12	12
Penta E	n.a.	13
D16S539	11	11
D18S51	n.a.	15
D2S1338	n.a.	25
CSF1PO	11/13	11/13
Penta D	n.a.	13
THO1	8/9	8/9
VWA	16	16
D21S11	n.a.	28/30
D7S820	8/10	8/10
D5S818	11/12	11/12
TPOX	8/10	8/10
D8S1179	n.a.	13/14
D12S391	n.a.	17/18
D19S433	n.a.	14
FGA	n.a.	21

Zelllinie PanC1		
Marker	Vergleichsprofil	PanC1
Amelogenin	n.a.	X
D3S1358	n.a.	17
D1S1656	n.a.	(12)/14
D6S1043	n.a.	11/12/(14)
D13S317	11	11
Penta E	n.a.	7/14
D16S539	11	11
D18S51	n.a.	12/(17)
D2S1338	n.a.	23/24
CSF1PO	10/12	10/12
Penta D	n.a.	14
THO1	7/8	7/8
VWA	15	15
D21S11	n.a.	28
D7S820	8/10	8/10
D5S818	11/13	11/13
TPOX	8/11	8/11
D8S1179	n.a.	14/15
D12S391	n.a.	22
D19S433	n.a.	11/(15)/16
FGA	n.a.	21

Zelllinie SiHA		
Marker	Vergleichsprofil	SiHA
Amelogenin	X	X
D3S1358	n.a.	16/17
D1S1656	n.a.	14/15
D6S1043	n.a.	18
D13S317	11	11
Penta E	n.a.	10/12
D16S539	12	12
D18S51	n.a.	15
D2S1338	n.a.	24
CSF1PO	12	12
Penta D	n.a.	9/12
THO1	6/9	6/9
VWA	14/17	14/17
D21S11	n.a.	29/(30)/31
D7S820	10	10
D5S818	9	9
TPOX	8	8
D8S1179	n.a.	13/16
D12S391	n.a.	19/22
D19S433	n.a.	14.2
FGA	n.a.	21

Zelllinie Ludlu1		
Marker	Vergleichsprofil	Ludlu1
Amelogenin	X	X
D3S1358	n.a.	15/17
D1S1656	n.a.	12/15/16/17.3
D6S1043	n.a.	10/11/12
D13S317	12	12/13/14
Penta E	n.a.	7/15
D16S539	13/14	9/11/13
D18S51	n.a.	16/17/18/19
D2S1338	n.a.	19
CSF1PO	11	10/11/12
Penta D	n.a.	9/10/13
THO1	6	7/9.3
VWA	17/19	15/19
D21S11	n.a.	27/28/29/30.2/32.2
D7S820	10/12	11/12
D5S818	12	8/9/11
TPOX	8/11	11
D8S1179	n.a.	12/13/15
D12S391	n.a.	19/21
D19S433	n.a.	14/17/18
FGA	n.a.	23

Zelllinie HeLa		
Marker	Vergleichsprofil	HeLa
Amelogenin	X	X
D3S1358	n.a.	15/18
D1S1656	n.a.	12/15
D6S1043	n.a.	18
D13S317	12/13.3	12/13.3
Penta E	n.a.	7/15
D16S539	9/10	9/10
D18S51	n.a.	(14)/16
D2S1338	n.a.	17
CSF1PO	9/10	9/10
Penta D	n.a.	8/15
THO1	7	7
VWA	16/18	16/18
D21S11	n.a.	27/28
D7S820	8/12	8/9/12
D5S818	11/12	11/12
TPOX	8/12	11/12
D8S1179	n.a.	12
D12S391	n.a.	20/25
D19S433	n.a.	13/14
FGA	n.a.	18/21

Zelllinie CaSki		
Marker	Vergleichsprofil	CaSki
Amelogenin	X	X
D3S1358	n.a.	15
D1S1656	n.a.	13/17.3
D6S1043	n.a.	11/14
D13S317	8/12	12
Penta E	n.a.	5
D16S539	11/12	11
D18S51	n.a.	17
D2S1338	n.a.	21
CSF1PO	10	10
Penta D	n.a.	10
THO1	7	7
VWA	17	17
D21S11	n.a.	30
D7S820	8/11	8/11
D5S818	13	13
TPOX	8	8
D8S1179	n.a.	13/(14)/15
D12S391	n.a.	18
D19S433	n.a.	15/16
FGA	n.a.	21

Zelllinie A431		
Marker	Vergleichsprofil	A431
Amelogenin	X	X
D3S1358	n.a.	14
D1S1656	n.a.	16/17
D6S1043	n.a.	18
D13S317	n.a.	9/13/(14)
Penta E	n.a.	12/13
D16S539	12/14	12/14/(15)
D18S51	n.a.	13/17
D2S1338	n.a.	17
CSF1PO	11/12	11/12
Penta D	n.a.	9/11
THO1	9	9
VWA	15/17	(13)/15/17
D21S11	n.a.	28/30
D7S820	10	10
D5S818	12/13	12/13
TPOX	11	11
D8S1179	n.a.	13
D12S391	n.a.	18/23
D19S433	n.a.	15/15.2
FGA	n.a.	20

2nd Editorial Decision

20th Jan 2020

Thank you for the submission of your revised manuscript to EMBO Molecular Medicine. We have now received the enclosed reports from the referees who were asked to re-assess it. As you will see, they are supportive of publication, and I am thus pleased to inform you that we will be able to accept your manuscript pending the following final editorial amendments.

***** Reviewer's comments *****

Referee #1 (Remarks for Author):

The authors have responded to all raised questions. No further comments

Referee #2 (Remarks for Author):

The authors have engaged well with the reviewer comments and should be congratulated for their thorough revisions and interesting new data. My previous comments have been addressed in the revised manuscript, which I believe should now be published.

Referee #3 (Comments on Novelty/Model System for Author):

No issues

Referee #3 (Remarks for Author):

The transcription factor Δ Np63 is a master regulator of epithelial cell identity and essential for the survival of squamous cell carcinoma (SCC) of lung, head and neck, oesophagus, cervix and skin. The authors here report that the deubiquitylase USP28 stabilizes Δ Np63 and maintains elevated Δ Np63 levels in SCC by counteracting its proteasome-mediated degradation. Impaired USP28 activity, either genetically or pharmacologically, abrogates the transcriptional identity and suppresses growth and survival of human SCC cells. CRISPR/Cas9-engineered in vivo mouse models establish that endogenous USP28 is strictly required for both induction and maintenance of lung SCC.

These data strongly suggest that targeting Δ Np63 abundance via inhibition of USP28 might constitute a promising strategy for the treatment of SCC tumours.

The paper has clinical relevance for SCC treatment. In this review cycle the authors have addressed my previous concerns and therefore significantly improved the quality of some data and overall quality of the manuscript.

2nd Revision - authors' response

5th Feb 2020

The authors made the requested editorial changes.

Corresponding Author Name: Dr. Markus E. Diefenbacher

Journal Submitted to: Embo Molecular Medicine

Manuscript Number: EMM-2019-11101-V2